# Mechanisms of Yellow Fever Transmission: Gleaning the Overlooked Records of Importance and Identifying Problems, Puzzles, Serious Issues, Surprises and Research Questions

**DOI:** 10.3390/v16010084

**Published:** 2024-01-04

**Authors:** Goro Kuno

**Affiliations:** Formerly at the Division of Vector-Borne Infectious Diseases, Centers for Disease Control and Prevention, Fort Collins, CO 80521, USA; gykuno@gmail.com

**Keywords:** yellow fever, transmission mechanism, history, biological transmission, arbovirus concept, reservoir, environment, viral dispersal, viral persistence, conventional wisdom

## Abstract

In viral disease research, few diseases can compete with yellow fever for the volume of literature, historical significance, richness of the topics and the amount of strong interest among both scientists and laypersons. While the major foci of viral disease research shifted to other more pressing new diseases in recent decades, many critically important basic tasks still remain unfinished for yellow fever. Some of the examples include the mechanisms of transmission, the process leading to outbreak occurrence, environmental factors, dispersal, and viral persistence in nature. In this review, these subjects are analyzed in depth, based on information not only in old but in modern literatures, to fill in blanks and to update the current understanding on these topics. As a result, many valuable facts, ideas, and other types of information that complement the present knowledge were discovered. Very serious questions about the validity of the arbovirus concept and some research practices were also identified. The characteristics of YFV and its pattern of transmission that make this virus unique among viruses transmitted by *Ae. aegypti* were also explored. Another emphasis was identification of research questions. The discovery of a few historical surprises was an unexpected benefit.

## 1. Introduction

Identified as the first human disease by a virus, yellow fever (YF) occupies a special place in history for its devastating impacts not only on public health but on economics. The research history of this disease is well known for the major milestones, sacrifices and triumphs, such as the discovery of the vector-borne mode of transmission, the development of a safe and highly effective vaccine, and the elimination of YF from the urban environment by means of thorough mosquito control. Those remarkable successes prompted many to apply a catchword “conquest of YF” in books and historical reviews. However, it was a premature selection, because many gaps in knowledge remain regarding the transmission mechanisms and environments. In fact, we still do not know how yellow fever virus (YFV) persists in nature, despite a considerable amount of research invested by many dedicated workers. These are the primary gaps in knowledge.

For some of the unresolved subjects, however, if one examines in depth the documents published in the past, it becomes evident that useful data was actually documented for certain topics but did not draw sufficient attention for the following reason. Whenever scientific documents are read for a review, the first reviewers choose the subjects based on their personal interest and judgement on the relevance or merit. Naturally, the subjects that fall out of the range of their interests or merit among contemporary researchers are often not presented. If this omission is repeated by later generations of reviewers, those subjects tend to be totally forgotten over time. Also, as the number of researchers increased and their range of interest diversified in modern times, the number or proportion of workers interested in far less popular subjects diminished, contributing to the further decline in interest in those forgotten subjects. 

However, many decades later, if the forgotten subjects suddenly become important topics, it is difficult to retrieve these early unrecognized documents, because the volume of literature would have grown enormously large by then. Thus, the modern day search becomes analogous to looking for a needle in a haystack. Though small in number, excellent reviews on YF transmission were actually published as book chapters in the past. Unfortunately, most of them have not been well read. Two possible reasons for the infrequent citation of book chapters are a trend of heavy dependence of researchers on computerized literature databases that do not cover sources other than periodicals and/or loss of interest in books in general. These problems have generated a “secondary kind of knowledge gaps”.

Fortunately, for some hidden documents, advancement in digital technology has made the task simpler, because an increasing number of academic and research institutions began posting old documents, which have been otherwise inaccessible for most researchers, online. Also, until recently, subject coverage in the reviews in journals was limited due to the prevalent practice of enforcing text size limits. However, such a limitation has been lifted recently in an increasing number of online periodicals because of the advancement in digital technology. Accordingly, presentation of a slightly lengthy but far more comprehensive review in periodicals is now possible and useful for drawing more attention to these neglected sources of valuable information. The combination of overlooked records of importance in the old records and review of the subjects from the modern literature is useful not only for updating the chosen subject but for filling the primary and secondary gaps in the understanding of transmission mechanisms or for identifying pertinent research questions. 

The first objective concerns the mechanisms of YFV transmission. In this review, the forgotten records or neglected subjects of importance discovered are brought to attention to supplement the current understanding of the mechanisms. The importance of the environment is emphasized to stress the uniqueness of YFV.

Historically, the discovery of YF as a disease transmitted by mosquitoes spawned arbovirology as a unique branch in virologic research; and intensive research on arbovirus transmission ultimately contributed to the establishment by the World Health Organization (WHO) of the foundation of the arbovirus concept based on biological transmission (BT) [1]. Though the WHO’s concept and research guidelines have generally been very useful, multiple serious problems were discovered. Despite these reports, puzzlingly, little has been addressed globally to revise the official concept. Accordingly, elaboration of the nature of the selected problems in the arbovirus concept and principles is the second objective.

Research on arbovirus transmission cannot be achieved in the field alone and requires laboratory experiments as an absolute necessity, because natural transmission mechanisms of any arbovirus are extremely complex. However, laboratory research has multiple limitations. In this review, some of the actual examples of miscellaneous research problems that have confronted researchers in the past are presented. This is the third objective.

Despite the popularity of YF, regarding some of the unresolved topics of considerable importance, it has been noticed that, for a reason totally unclear, either only insufficient amounts of background information have been provided or few attempts have been made to even propose a hypothesis. However, when one pays closer attention to these topics in the literature, depending on subject, it becomes evident that a lot of data of considerable significance have remained uncovered for many decades. Thus, this review is designed to explore those topics more in depth by collating numerous sources of information in order, analyzing the significance, and, where necessary, synthesizing new hypotheses through mobilization of available data, to induce interest in those somewhat neglected subjects of importance. 

Except in the sections concerning the arbovirus concept, selected problems encountered in research and comparison of the characteristics among four viruses transmitted by *Ae. aegypti* in search of the uniqueness of YFV, the data covered in this review are largely about YF and YFV. It is emphasized that, in this review, original sources are listed as much as possible. This is necessary because misquotation of sources through secondary reading has proliferated in modern times, resulting in occasional unfortunate inaccuracy. Also, giving due credit to the original authors is still important in science.

## 2. The Evolution of the Concept of the Vector-Borne Transmission of Yellow Fever and Other Arboviral Diseases

### 2.1. A History of the Discovery of the Vector-Borne Transmission of YF

Prior to the ground-breaking discovery by the U.S. Army’s YF Commission led by Walter Reed (hereafter called “Reed Commission”), for nearly three centuries, fierce battles were waged over the cause and mode of transmission of YF, pitting contagionists against anti-contagionists in Europe and North America. The YF outbreak in Barcelona in 1821 was a memorable occasion for major public debate. Troubled by the lack of consensus, one of the renowned French physicians at the time, Nicholas Chervin, even visited the YF-afflicted ports in the U.S. in 1820–1821 to collect different ideas on the cause and peculiarities of the YF outbreak. It was a long seesaw battle because the dominant side shifted back and forth multiple times until the 1860s [2]. A retrospective analysis after the discovery of mosquito-borne transmission revealed that, ironically, both sides were partially correct [3]. 

For a long time, it was understood that the idea that the mode of the transmission of YF involved insects began to evolve towards the end of the 18th century, culminating in the legendary discovery of the role of mosquitoes by Carlos J. Finlay in 1881 and the definitive confirmation by the Reed Commission in 1900. Though Finlay’s contribution is the best known, even among many laypersons, contributions of other earlier workers deserve more attention. 

John Crawford (1746–1813): He was a former Irish businessman turned physician who became prominent in Baltimore, U.S. He considered quarantine for YF as an exercise in futility and, instead, proposed the involvement of insects in the causation of YF, an idea he developed while working in Dutch Guiana, in 1807. He was ridiculed for his unusual proposal and ostracized not only in the medical community but in the general public. He suffered financially, because patients stopped coming to his clinic, influenced by the bad publicity [4]. Nevertheless, he still managed to publish his conviction in a strongly worded appeal to the medical community to reconsider nearly 70 years before Finlay [5]. Though he was the first to propose the insect theory for YF among the early proponents and his contribution was known to some prominent YF workers around the turn of the 20th century, unfortunately he has been the least recognized in history. 

Josiah C. Nott (1804–1873): He was a physician in Alabama, U.S. He strongly suspected the involvement of animalcules carried by insects in the cause of YF and published his opinion in 1848, though he could not explain the exact mechanism. By sheer coincidence, when William C. Gorgas was born, Nott was the attending physician to his mother. Nott’s interest and conviction on the cause were not only scientific but strongly personal because he lost four of his children to YF in one week [6].

Louis D. Beauperthuy (1807–1871): He was a French physician who, while working in Venezuela, proposed the involvement of mosquitoes in the cause of YF in 1854 [7]. He suspected that the mosquitoes picked up something poisonous from the putrefying soil around the house and somehow transmitted it to the residents [8,9,10]. Interestingly, long after his death, the mosquito specimens he had collected and which he called “striped-legged mosquito” were discovered and definitively identified as *Ae. aegypti*.

Carlos J. Finlay (1833–1915): He was a Cuban physician medically educated in the U.S. but who had training in Europe as well. His theory of mosquito transmission published in 1881 [11] was pivotal in the historic success of the Reed Commission in 1900. Nearly all biographers of Finlay and others interested in the subject at the time stressed the originality of the mosquito concept by Finlay. However, at around the turn of the 20th century, multiple YF investigators (such as Henry R. Carter, Aristides Agramonte, and Rubert Boyce) were aware of the contributions by Beauperthuy, Nott, and a few others. This raised a question as to how Finlay could remain unaware of them. François Delaporte was a French historian and philosopher who became interested in the genesis of Finlay’s ground-breaking idea. After an exhaustive archival examination in Cuba, U.S. and Europe and interviews with the individuals who were knowledgeable about Finlay’s work, he concluded that it was impossible for Finlay to conceive the idea of mosquito transmission without reading Patrick Manson’s publication [12,13]. 

Aristides Agramonte was born in a battle field in Cuba when his father (a physician) fought for Cuban independence as a general. As his family moved to the U.S. after his father’s battle field death, he was educated there and appointed as a member of the Reed Commission. Nonetheless, for Finlay, Agramonte was still a Cuban compatriot and a close friend. In an article, after making references to the earlier discoveries of Beauperthuy and Manson, Agramonte stated that “Manson had already demonstrated what the insect could do in transmitting disease since 1870. Until Manson’s demonstration of filaria transmission, since Beauperthuy’s articles (between 1854 and 1870), no mention of insect agency in the propagation of diseases can be found in medical literature. Finlay quickly followed, in 1881, with a paper read before the Havana Academy of Sciences…” [9], ambiguously hinting that Finlay was aware of the earlier works of both Beauperthuy and Manson.

### 2.2. A Brief History of the Establishment of the Arbovirus Concept Based on Biological Transmission (BT)

As well known in history, the success of the Reed Commission depended on adoption of Carter’s crucial discovery of the “extrinsic incubation period” (EIP), a requirement for the transmission of the YF agent by the mosquito, which Finlay had incriminated earlier. 

Unrecognized in the YF research history for more than two centuries, actually it was David Hosack (1769–1835) who recognized for the first time in 1814 a peculiar interval of several to 14 days of normalcy (no occurrence of new YF cases) between eruptions of clusters of YF patients [14]. Hosack was a physician in New York who had multiple opportunities to study the pattern of YF spread, since the city was hit by YF 15 times. He was a student of Benjamin Rush, with whom he later disagreed regarding the cause of YF. He was intrigued by the puzzling interval of no YF reporting in the midst of outbreak, because many patients in one cluster and other patients in the following cluster lived in the same neighborhood. He noticed that when YF was introduced to port cities in the eastern seaboard of the United States, its spread was initially very slow and gradual. But, thereafter, clusters of YF patients were separated by this peculiar interval of no new patient being recorded. Hosack consulted with physicians in other port cities hit hard by YF and learned that some of the physicians in those port cities also recognized this interval of no YF case. However, unlike those physicians, Hosack speculated a relationship between YF outbreaks and the abundance of mosquitoes [15]. Of course, at that time, Hosack, unlike Carter nearly 84 years later, did not know anything about the relationship of the interval without new YF patients and the EIP. Unquestionably, Hosack knew nothing about the name of the mosquitoes either.

Carter was a highly dedicated and meticulous physician of the U.S. Public Health Service. In 1898, while working in the rural communities along the Mississippi River to investigate the mode of YF transmission, he noticed an approximately 14-day interval between clusters of YF patients, which he named the “extrinsic incubation period” (EIP) [16,17]. Apparently, Carter was totally unaware of Hosack’s earlier discovery. As mentioned above, integration of this crucial information to Finlay’s mosquito theory was indispensable for the success of the Reed Commission. Carter’s discovery was quickly confirmed by French workers in Rio de Janeiro, Brazil [18]. 

After the first isolation of YFV by the investigators of the Rockefeller Foundation (RF) in 1927, laboratory experiments firmly established a mosquito-borne viral transmission. This led to more discoveries of vector-borne transmission; and by 1930, there were six such viruses. Shortly, Karl F. Meyer laid out the foundation of this mode of viral transmission [19]. Since Meyer, a Swiss-born microbiologist who emigrated to the U.S. via South Africa, was a firm believer of the zoonosis concept originally developed by Rudolph Virchow, the latently infected vertebrate reservoir was central to vector-borne viral transmission. His isolation for the first time of Western equine encephalitis virus from a dead horse in California strengthened his conviction of the zoonotic concept. It is emphasized that while Meyer laid out the basic framework of the arbovirus transmission scheme, because he wished to continue research in bacterial diseases (the subjects of his passion), he left many gaps in knowledge in the transmission cycle to be filled by the later generations of workers. Shortly, the term “biological transmission (BT)” was adopted for arboviral transmission by William C. Reeves, a graduate student of Meyer. Actually, this term was borrowed from the cyclo-propagative mode of vector-borne plant virus transmission recognized by Karl Maramorosch in plant pathology. It was Reeves who coined the term “arbovirus” as a sigla for “arthropod-borne virus” at the International Congress of Tropical Medicine and Malariology held in Lisbon, Portugal in 1958 [20]. The term was officially approved by the WHO in 1963. 

The WHO subsequently published an official arbovirus concept which was largely based on Meyer’s concept [1]. Because arbovirology was still evolving and many gaps had not been filled, guidelines and other forms of ideas to avoid pitfalls and to assist researchers to proceed to successful research were integrated in this WHO’s concept. The WHO published a revised but more abridged version in 1985 [21]. But, because basically the WHO did not change much and left the concept virtually intact, so the 1967 version is used for all discussions in this review. 

## 3. Vectors

In the early part of YF history, arthropods involved in transmission to humans were called “intermediate hosts” (equivalent to “mere carriers”). In retrospect, when Finlay was using mosquitoes in his studies prior to 1900, he was trying to prove mechanical transmission, rather than BT, of the putative YF agent. 

Long after the Reed Commission’s study in 1900, Richard M. Taylor classified vectors into three categories: dead-end vectors”, “possible vectors”, and “true or natural vectors” [22]. In reality, such a subdivision was not practical or accurate because most virus–vector interactions in nature overlap. After the discovery of *Ae. bromeliae* as a vector of YF in the transitional environments (Section 5.4.2), “bridge vector” became another term popularly used. However, proliferation of the broader application of the term “vector” to other types of animal viruses, such as its questionable application to rodent viruses as “rodent-borne viruses”, became a new source of problems, necessitating reappraisal of the definition of “vector” [23]. It is emphasized here that all vectors of importance must blood-feed at least on two occasions on primary hosts in nature to ensure a successful BT, first by normal vectors to acquire the virus from a viremic host and second by infective vectors to transmit the virus to uninfected host. 

### 3.1. Ae. aegypti

This so-called “yellow fever mosquito” has had a long history of taxonomic problems [24]. After the discovery of *Ae. albopictus* in Texas in 1985, the Asian colloquial name, “tiger mosquito” [25] was adopted as “Asian tiger mosquito”. However, well before 1911, “tiger mosquito” was used for *Ae. aegypti* in British colonies [26]; however, in Asia, “white-striped mosquito” was the local name of *Ae. albopictus*.

By the time of Finlay’s discovery of mosquito-borne transmission, this mosquito was well recognized around the world for its global distribution, ease of rearing, and convenient source of material in biological experiments. Thus, when the Reed Commission was investigating YF in Cuba, textbooks on mosquito rearing and natural history were already available [27]. Also, because of the long survival (for more than 3 months and as long as 1889 days [28]) of the eggs dried on paper, the eggs were exchanged by mail among YF researchers in Africa, the Caribbean, South America, Europe and North America. Thus, the identity of African, Asian and New World strains was established [29]. A laboratory colony known by “ROCK” kept at the Rockefeller Institute of Medical Research (RIMR) in New York was originally established from the eggs mailed from Wilhelm H. Hoffmann. He was a German naval physician who specialized in YF and who was appointed to succeed Carlos Finlay as Director of the Finlay Institute in Havana, Cuba upon Finlay’s death in 1915 [30,31]. Thus, it is the oldest laboratory colony of *Ae. aegypti* maintained in the world. The first comprehensive listing of the geographic distribution of this mosquito and sylvan vectors around the world was made available in 1931 by Henry W. Kumm. Interestingly, he listed *Ae. albopictus* as one of the potential vectors of YFV, with its distribution widely in Asia but exceptionally also in Djibouti of Africa as early as 1909 [32].

Global dispersal: Phylogenetic studies largely confirmed the hypothesis that this mosquito originated in East Africa as a sylvan subspecies (*Ae. aegypti formosus* or *Aaf*) feeding on nonhuman primates (NHPs). After its dispersal to West Africa, a subspecies (*Ae. aegypti aegypti* or *Aaa*) adapted to the urban environment evolved. According to a recent study, domestication apparently took place somewhere in Western Sahel approximately 5000 years ago [33]. While subsequent dispersal of the domesticated lineage from West Africa to the Americas in association with the slave trade has been well supported by phylogenetic studies, how widespread the *Aaa* form is in West Africa today has been a controversial topic because of the difference in tree topology depending on the gene sequenced. Actually, the practice of recognizing two subspecies of *Ae. aegypti* has become a subject of controversy lately [34]. Also, contrary to the traditional understanding, a recent report from northern Ghana revealed that 31% of man-biting *Ae. aegypti* are *Aaf*, strongly suggesting the involvement of this subspecies in urban YFV transmission [35]. The route to the further dispersal of *Aaa* from the Americas to Asia and the Pacific has not been well established either, though dispersal to Asia via Suez Canal in 1869 has been a leading theory [36]. The update of this question is found in Section 3.6.

Bionomics: The monumental compendium of the natural history and other biologic data of this mosquito by S. Rickard Christophers [37] has served as an invaluable source of basic information on this mosquito. A small number of more recent exceptional findings are added here to complement the above compendium. 

Although Finlay claimed that this mosquito did not occur at a high altitude, the mosquito has been known to occur in Mexico City (at an altitude of 1700 m above sea level) even in Finlay’s time. More recently, it was found indoors, even at an altitude of 2300 m; and dengue virus was isolated at 1984 m above sea level in Colombia [38]. The sylvan subspecies in Africa breeds in plant axils and tree holes and is zoophilic, while the urban subspecies is domestic. Its house-entering behavior was reported to be determined genetically [39]. It was Raymond C. Shannon who proposed for the first time the theory that *Aedes* mosquitoes evolved from breeding in tree holes to container breeding in human-modified environments [40]. Existence of intermediate subpopulations demonstrating exophilic (outdoor) feeding on NHPs but only modest feeding on humans is also recognized in Africa [41]. In Réunion Island in the Indian Ocean, two vectors (endophlic or indoor *Ae. aegypti* and exophilic or outdoor *Ae. albopictus*) reversed their typical habitat, the former becoming exophilic and the latter becoming endophilic over decades [42]. When an extensive chikungunya (CHIK) outbreak occurred in the Réunion Island of the Indian Ocean in 2005, emergence of a genetic mutant of chikungunya virus (CHIKV; E1-A226V) adapted to *Ae. albopictus* was popularly featured as the cause in most publications. However, the significance of this earlier outdoor-indoor switch history between the two vectors there has been rarely mentioned.

This mosquito is well known for breeding in clean water in household containers worldwide. However, it was known already in the very early part of the 20th century that, depending on places in Arica, local subpopulations had adapted to breeding in turbulent or septic water [43].

This was more recently reported from Africa and the Caribbean [44,45]. Its diverse larval breeding habitat also includes underground water reservoir [46] as well as rock holes. Although this species is generally diurnal biters, a subpopulation in parts of Ivory Coast are nocturnal biters, feeding activity peaking at midnight [47]. In Trinidad, 10% of this vector feed at night [48]. Its multiple feeding enhances virus transmission, but even its partial feeding is known to be sufficient in transmitting virus. As for longevity of adult females, though highly unusual, they survive a maximum of 2 months without food. Carter also learned that nearly 50% of females survive for as long as 101 days and one as long as 154 days without food [49]. Flight range also varies, depending on the design of experiment. If human dwellings are near the release point, most of them travel much less than 100 m. As George C. Bugher proved by releasing radio-labeled mosquitoes, if released in a natural environment without a human habitation, this mosquito would travel as long as 1270 m [50]. 

Generally, *Aaf* is less susceptible to YFV infection compared with *Aaa*. As mentioned above, its taxonomic segregation throughout Africa was questioned lately. In rural communities in East Africa and urban centers of Central Africa (i.e., Angola), both *Aaf* and *Aaa* coexist. How this coexistence affects the intensity or duration of YF transmission has not been sufficiently studied.

### 3.2. The Extrinsic Incubation Period (EIP)

The EIP has been largely determined in the past using *Ae. aegypti*. In Carter’s hands, the EIP was approximately 14 days [17]. Temperature is known to impact on the EIP. Generally, the higher is ambient temperature above 16.5 °C, shorter is EIP. However, when temperature exceeds 35–36 °C, longevity of the mosquito drops. In one study, at 25 °C, it was 12–16 days, while at 35 °C, it was 7.3 days [51]. It is also known that virus strain, infectious dose, and mosquito strain used in studies contribute to variation in the EIP. In a sylvan vector in the Americas, *Haemagogus capricornii*, at 25.4 °C, it was 20–24 days, while at 30 °C, it was 13–15 days [52]. 

### 3.3. Vertical, Venereal and Direct Modes of Transmission in Vectors

Interest in vertical transmission (VT) of YFV arose soon after the confirmation of vector-borne YFV transmission in 1900. A team of workers from the Institut Pasteur reported an observation of VT of YFV in *Ae. aegypti* mosquitoes [18]. However, all attempts to confirm it by numerous investigators thereafter failed, until it was confirmed under natural conditions nearly 95 years later [53]. VT is almost synonymous with hereditary or transovarial transmission (TOT), although trans-ovum transmission is another form of VT. The venereal transmission of YFV in *Ae. aegypti* was also reported [54]. 

The main objective for conducting VT experiments in the early part of history was to collect evidence to support the idea that it was the mechanism of YFV persistence in urban transmission. As no evidence of YF persistence in urban areas was obtained and viral persistence in an enzootic/epizootic environment became far more important, the interest shifted to VT in sylvan vectors. In a study in the Brazilian Amazon, as much as 2.9% nulliparous females of *He. janthinomys* were found infected with YFV [55]. In other study, however, evidence of TOT in sylvan vectors could not be obtained [56]. In Africa, YFV was detected from males of *Ae. furcifer-taylori* in Sénégal [57]. However, generally, in both the Americas and Africa, it has been difficult to obtain evidence of TOT in sylvan vectors. While infrequent occurrence is a possibility, one of the other suspected reasons for the difficulty of studying TOT is the constant shift in enzootic/epizootic foci. As a result, after a site for research in sylvan environments is selected, by the time sampling begins, the active foci would have moved away already. Generally, it is appropriate to consider the mechanism of viral persistence as multifactorial, with VT being only one of them.

A well-known example of direct transmission (DT) is the *Ae. aegypti* larval transmission of YFV in an aquatic environment [58]. Despite its potential importance in the persistence and transmission of the virus in nature, not much has been done thus far. The other DT mechanism is expectoration. It is known that when *Ae. aegypti* males and females engage in nectar feeding outdoors, they share the nectar sources [59]. As occurs in blood feeding, some salivary materials are discharged into the nectar during feeding. Accordingly, it is possible that YFV-infected females discharge YFV into nectar, which then infects other uninfected females or males that imbibe the contaminated nectar. Thus, like in larval transmission, it may be significant for YFV transmission or persistence in enzootic foci in sylvan environments. However, very little has been investigated thus far.

### 3.4. Vectors as Reservoirs

The word “reservoir” has been applied as a collective word to all qualified organisms as a whole, a physical space where a pathogen is maintained naturally, or in reference to particular species or to each member of a species. In this review, the first and third usages are applied. 

Since existence of a reservoir for YFV was a dominant interest in the early part of history, it was one of the major objectives of the RF which conducted YF research (probably the most expensive and exhaustive field research on any single infectious disease at the time) in two continents for 24 years. Accordingly, research by early workers was mainly focused on finding vertebrate reservoirs. The investigators examined a large number of hosts and invertebrates for evidence. Despite the enormous investment, the RF staff, as well as French workers, concluded that true reservoirs of YFV were vectors but not hosts [22,50,60,61,62]. Actually, skepticism about a host reservoir for YFV has a long history, beginning with a flat denial of infected humans as reservoirs by Carter [63]. More negative data are presented in Section 4.5. Behind this puzzling adoption of the vertebrate reservoir in arbovirology lay a complicated history [64]. Interestingly, even the architect of the arbovirus concept, Meyer, recognized asymptomatic infection of *Ae. aegypti* as an example of reservoir status [65]. Identifying vectors as true reservoirs of YFV and many other arboviruses, however, requires establishment of at least two new universal definitions, one for mosquito vectors and another for tick vectors. This is because the arbovirus concept was built mostly on the biology of mosquito vectors. As an example, the intervals between blood-feedings of ticks are very long and unpredictable due to their dependency on chance encounters with a host. Thus, the concept of the EIP cannot be directly applied to tick-borne viral transmission. Furthermore, a blood-feeding stage in the life cycle of ticks and the process of blood digestion are also different. Thus, it was not surprising that a protest was lodged that ticks are not another group of mosquitoes [66].

Another consideration for the evidence of the vector’s role as a reservoir is long survival of certain sylvan vectors (such as *Ae. luteocephalus* in Burkina Faso and *Sabethes chloropterus* in Central America) during dry seasons [67,68]. In fact, long persistence of YFV RNA was detected during dry seasons in sylvan vectors in the Americas [69].

### 3.5. The Minimal Size of the Vector Population Necessary for Viral Persistence

At the time when YF was thought to be an urban disease transmitted by *Ae. aegypti* in “endemic centers”, Carter believed that YF would spontaneously disappear if the population of infected vectors in a given area was brought to a level below “minimal”, a concept originally conceived by Ronald Ross for eliminating malarial transmission [63,70]. According to Carter, because immunity to YF is permanent for the survivors, if the size of the vector population was reduced to a level below threshold, one could eliminate YF spontaneously. Carter applied a 1:1 contact ratio between vectors and susceptible humans. Because *Ae. aegypti* mosquitoes engage in multiple feedings and even partial feeding (which is often sufficient for transmitting virus), Carter’s proposal was not received well. However, at least as far as controlling epidemic in the urban environment is concerned, it was stressed that minimizing the number of infective mosquitoes in the vector population was a key [68]. 

### 3.6. The Transoceanic or Transcontinental Dispersal of Vectors

This topic only concerns *Ae. aegypti.* The dispersal of *Ae. aegypti* from West Africa to the Americas in association with the slave trade [71] has been strongly supported by phylogenetic studies. However, well before the age of molecular virology, Carter came to the same conclusion based on reading archival documents [49]. 

During the height of frequent YF outbreaks in North America and Europe in the 17–19th centuries, sailing vessels played a major role in transporting YFV-infected *Ae. aegypti* mosquitoes and/or infected crew or passengers. Because sailing ships were not mechanized, they required a large crew for loading and unloading of goods. As these ships travelled very slowly, depending on the route, the journey often took as long as two months or longer from South America to the destinations. They also had to anchor at more ports during travel for a resupply. Most importantly, because YF was prevalent not only at the ports of provenance but at some of those stopovers, this long journey provided a perfect condition for YF transmission to occur on board ships. The crew who disembarked could be infected at the port but boarded again. When passengers or crew boarded ships at stopovers, *Ae. aegypti*, which followed them very closely behind, also boarded. Furthermore, many water barrels or kegs were required on deck, an ideal condition for mosquito breeding. 

On the other hand, steam ships were more mechanized and travelled faster, thus requiring fewer crew and often no stoppage for a resupply. Also, no water barrel was necessary because of the concealed water tank and piped water system. Thus, the opportunities for mosquito breeding and importation of the YF agent were drastically reduced. Also, at that time, in North America and Europe, sugar trade with the Caribbean and South American regions declined because the source of sugar could be found elsewhere by that time. This is reflected in the sharp decline in the number of YF outbreaks in areas with a temperate climate in the late 19th century. However, even steamers were involved in a far smaller number of outbreaks.

In the heyday of sailing ships, the quarantine service in the U.S. conducted a thorough inspection of the ships arriving from YF-endemic locations. Ship Island off the coast of Mississippi was one of the inspection stations. Ships were required to anchor at a designated port and at a safe distance (more than the flight range of mosquitoes) offshore. Log books of the physicians on board were inspected and passenger manifest was checked for any indication of YF. Selected passengers were further interviewed intensively. Because Carter spent a huge amount of time on inspection at multiple quarantine stations, he acquired a very comprehensive knowledge about the mechanism of YF dispersal by ship. He found out that adult *Ae. aegypti* survive as long as 154 days and also confirmed observation of the EIP (14 days or over) even onboard ships [49]. As to the geographic route of the dispersal of *Ae. aegypti* to Asia and the Pacific, Carter definitively determined 1826 as the date of the introduction of this mosquito to Hawaii (a U.S. territory that was free from this mosquito until then) by a ship from San Blas, Mexico, based on the records of the U.S. Quarantine Office in Hawaii [49]. 

By the early 1930s, aircraft began to replace steamships in commerce and passenger transportation. Then, the discovery of *Ae. aegypti* in the cabin of aircraft arriving from tropical locations became a new concern [72]. Thus, aircraft inspection and passenger examination became necessary to prevent the introduction or spread of YF to Asia [73]. In the British colonies in Africa, not only aircraft but train, boat and automobile became the targets of investigation, as suspicion over their role in YF dispersal arose [74]. 

Though spread of this vector by human activities was also a shared concern in dengue research, an interesting distinction of dengue from YF has been recognized. In Australia, a dengue outbreak occurred in a gold mine shaft infested by this mosquito in 1885 [75]. This happened because *Ae. aegypti* follows humans who move very closely for a reason. Thus, in urban areas that suffered a dengue outbreak, human movement from house to house was identified as a mechanism of viral spread [76]. Also, dengue outbreaks followed very quickly after the dispersal of *Ae. aegypti* to new territories by commercial air services. As an example, in the 1988–1989 dengue epidemic in French Polynesia, the island that experienced the earliest outbreak corresponded to the destination with the highest volume of passengers and the highest number of flights [77]. It is noted that despite the frequent occurrence of *Ae. aegypti* dispersal to virgin territories, unlike dengue, YF spread has not followed outside the traditional YF zones in Africa and the Americas.

Though rarely mentioned in the literature, the fact that none of the major sylvan vectors in Africa and Americas have ever dispersed outside the YF zones is very important when evaluating the possibility of the future dispersal of YF. As for minor YF vectors of Africa, *Ae. vittatus* is known to occur in Indian Ocean islands and widely in warm regions of Asia, while *Ae. bromeliae* is found in Madagascar and elsewhere in Africa as well as in Comoros. But these vectors are of secondary importance.

### 3.7. The Vectorial Capacity and Vector Competence

The origin of the concept of measuring the transmission capability of invertebrate vectors is traced back to Ross, who designed a mathematical formula in 1915 for the condition that favored spread of malaria [78]. Ross called it the “vectorial capacity”. In this review, this term is used to encompass both intrinsic factors (i.e., the minimal viremia level necessary for transmission, the minimal size of the vector population, etc.) and extrinsic factors (i.e., population size or density of vectors and hosts, probability of the meeting among virus, kind of vectors and of hosts, climatologic and environmental conditions favorable to virus transmission, etc.). This concept was further advanced by George MacDonald [79]. 

As the concept gained popularity, variations in the understanding of this concept evolved. The concept of vector competence is one such variation. Historically, it was Jean Legendre who advanced a theory that there exist multiple “races” of *Stegomyia fasciata* (now *Ae. aegypti*) in terms of the capacity to transmit the virus [80]. This stimulated others to conduct an experiment to evaluate and compare the efficacy of YFV transmission among the subpopulations of this mosquito in tropical and subtropical areas of the world [81,82,83]. The main objective was to answer why YF has never occurred in Asia [84]. These pioneers, however, did not use the term “vector competence”, which was coined much later. However, by the early 1930s, this type of study had become highly popular. Edward Hindle alone examined the transmission capacity of 38 species of mosquitoes for YF [85]. After 1950, the application of the vector competence test with respect to this Asian puzzle became popular [86,87,88]. 

According to some, “vector competence” is mostly used for a laboratory simulation using a smaller number of parameters pertaining to BT, since integration of most environmental parameters is impossible. Roy W. Chamberlain and his colleagues thought that vector competence was a genetically fixed intrinsic trait. Thus, they attempted to quantify the virus–vector relationship by emphasizing the threshold of infection, the infection rate, and the transmission rate [89]. 

James L. Hardy and his colleagues studied the midgut barrier, dissemination of viral infection to other organs, salivary gland infection, and others. Furthermore, they firmly believed that vector competence is a more dynamic and broader concept. Thus, in their minds, vector competence was a product of complex interactions among many parameters, including both intrinsic and extrinsic parameters (including variables such as quantitative changes in the microenvironment, the population dynamics of vectors and hosts, spatiotemporal parameters regulating the meeting among the virus, the vector and the host and others). Variations in vector and/or host population size, as well as its dynamic change over time, were also important because it impacts on synchrony among the major players of BT. They also considered the involvement of interference mechanism within vector. Thus, in their minds, vector competence was mostly synonymous with the vectorial capacity [90,91].

Definition, limitation, and related problems: The absence of a universally agreeable definition of vector competence and the synonymous application of “vector competence” and the “vectorial capacity” were the two sources of major confusion. As each group used a custom-made experimental protocol, the lack of standardization was another issue [92]. As examples, how artificial blood containing virus is prepared and what feeding device is used affect the result considerably. More recently, one commercial feeding device has become popular, thus reducing variations. One lesson many investigators learned quickly was to not use freshly thawed virus stocks. The criteria used to differentiate “competent” from “incompetent” are subjective, because they depend on personal judgment. All of these variations have made comparing results among groups of workers for the same vector and/or virus difficult. In one report on the *Ae. aegypti* strain colony established from a location in Africa, where an extensive YF outbreak had occurred shortly before, the colony was found vector incompetent [93]. These results were interpreted to show the limitation of the vector competence that cannot integrate multiple environmental factors. 

The increase in the number of parameters used in vector competence studies has been another factor complicating the definition of vector competence. For example, as molecular biologic techniques advanced, previously unknown antiviral defense mechanisms in vectors and existence of co-inhabiting viruses or bacteria were found to affect vector competence. 

As for YF, nearly all vector competence studies have been performed with *Ae. aegypti* strains. This mosquito is still important as long as the urban transmission of YF is the theme of study. However, this test should be performed more often on sylvan vectors, since these vectors provide the crucially important information about the mechanisms of sylvan enzootic and YFV persistence. In contrast, similar studies with *Ae. aegypti* do not yield a relevant answer to these questions. Clearly, the difficulty of establishing a laboratory colony of sylvan vectors has been the major source of the lack of progress.

## 4. Hosts

### 4.1. Human

After the breakthrough discovery of YF transmission by *Ae. aegypti*, it was initially thought that humans were the only host of this agent [94]. As well known, in the very early period of YF investigation, human volunteers were used in experiments. The tragic deaths and ethical issues that arose prompted Reed to propose a prototype “informed consent form” [95]. After more refinement, informed consent became a standard of medical ethics. As for the role in urban transmission, though most of them are incidental hosts, some of them serve as amplifying hosts.

### 4.2. Nonhuman Primates (NHPs)

Actually, the interest in the hosts of YFV evolved before humans were identified as primary hosts in urban transmission. The puzzling coincidence between YF outbreaks and the death of howler monkeys was recognized by Charles Kingsley in Trinidad during an outbreak of YF in 1869 [96]. Of course, he did not know what the death of the monkey meant at the time. On the other hand, when Andrew Balfour learned of the same, intriguing information (massive death of howler monkey, *Alouatta seniculus*) during his visit to Trinidad in 1914 [97], the time was ripe for speculating its relation to YF in NHPs. Besides the speculation of Balfour, similar suspicions raised by multiple others (including Harald W. Thomas and Manson) further increased the interest in this connection [98]. However, the initial British attempts to use howler monkeys for experiments failed because many neotropical primates could not adapt to the conditions in captivity. The crucially important suggestion of Oskar Klotz to use Asian monkeys instead led to the first isolation of YFV and the role of NHPs as hosts of YFV [99,100,101]. 

With the discovery of the sylvan transmission of YFV in South America, attention shifted to nonhuman hosts in rural and sylvan environments. Nelson C. Davis was one of the earliest workers who confirmed YFV replication in howler monkeys [102]. Many other species of NHP were then found to be the primary hosts in sylvan environments in the Americas and Africa. Currently, NHPs serve as amplifying hosts in the BT of YFV.

### 4.3. Incidental Hosts

This group is also called dead-end hosts, tangential hosts or accidental hosts, depending on the author. All types of hosts which do not contribute to BT after the bite of virus-infected vectors fall in this category [1]. They are the consequences of opportunistic feeding of sylvan vectors of YFV. These vectors, unlike *Ae. aegypti* in urban settings that feed almost exclusively on humans, tend to engage in indiscriminate feeding depending on the availability of a blood meal source. The sources dynamically change geographically and temporarily. In reality, it is difficult to discriminate incidental hosts from secondary or tertiary hosts of YFV. Nonetheless, it is generally recognized in South America that birds, bovines, rodents, opossums, and others fall in this category of hosts for *Haemagogus* vectors. In YF transmission in Africa, cow, horse, sheep, goat, pig, camel, dog and others demonstrating high antibody seropositivity to YFV were incidental hosts [74]. For *Ae. furcifer* and *Ae. taylori* in Africa, incidental hosts are often bats, rodents, and birds. However, the host membership of this group varies considerably from place to place.

There evolved a concept that an abundance of incidental hosts reduces the efficacy of viral transmission through a dilution effect. On the other hand, because of the lack of interest in incidental hosts, the hidden role of these hosts has not received a sufficient amount of attention. This relates to the benefit to vectors. Nearly 90 years ago, Johannes E. Dinger discovered that once *Ae. aegypti* females are infected with YFV, subsequent feeding on blood containing antibodies to the virus acquired from immune humans does not eliminate infectious virus in the vector. Davis concurred with this finding shortly [103,104]. Although it is not known well if similar studies on the sylvan vectors of YFV have been performed, it is almost certain that these sylvan vectors are no different from *Ae. aegypti* in this regard. If so, those infected females may still be able to transmit the virus in the next opportunities by feeding on uninfected hosts they encounter. Since many YFV vectors are multiple feeders, this scenario is highly probable. Meanwhile, blood meals from immunized incidental hosts are actually beneficial for the growth of the offspring of the mosquito by providing necessary nutrients. Accordingly, some (but not all) of the incidental hosts indirectly assist BT. This may partially explain why YFV persists in enzootic foci despite very high seropositivity in the NHPs. 

### 4.4. Viremia

All other factors being equal, the higher the viremic level and the longer the viremic period, the more efficient the BT. In humans, YFV viremia starts 2–3 days after infection but before the onset of illness and may last for 3–5 days. As variations in host factors among patients and genetic factors among virus strains are common, YFV could be isolated from a patient as late as 12 days after the onset of illness, though such a case is uncommon [105]. After vaccination, mild viremia of the vaccine strain may occur for 3–7 days, depending on vaccine recipients [106]. Viremia in NHPs may last from a few to 7 days. Interestingly, under laboratory conditions, the duration of viremia in NHPs is often inversely related to the viral concentration in the inoculum [50].

For the titration of the viremia level in blood, initially, the dilution endpoint was used. However, the method is subjective because the titer depends on the original concentration of the virus. To improve the titration method, the lethal dose 50% (LD_50_) method was developed for YFV by Hugo Muench [107]. 

Then, the plaque assay was discovered as an excellent method of virus titration. The first animal virus plaque assay was developed by Renato Dulbecco (later Nobel Laureate) not with YFV but with Western equine encephalomyelitis virus [108]. 

Two major issues surrounding the measurement of viremia need to be addressed. First, the transmission of some arboviruses without evidence of viremia [109,110,111]. While extravascular virus as a source of vectorial infection in those reports is a possible explanation for tick-borne viruses, the other cause of this observation is the insensitivity of the techniques used for detecting the infectious virus. Among the virus isolation methods available, it has been recognized that the intrathoracic inoculation of adult mosquitoes [112] is by far the most sensitive among the techniques available, including isolation in mosquito cell cultures and the intracranial inoculation of suckling mice. Historically, this method was first developed by parasitologists who experimented to confirm the transmission of helminthic parasites by mosquitoes. Then, it was applied to study plant virus transmission by leafhoppers and other insects. However, the application of this technique has been discouraged lately because the technique is too cumbersome to perform and because it requires experience [113]. Accordingly, it is highly probable that some of the culture-negative blood specimens could have been classified as viremic had the mosquito inoculation technique been applied. 

Second, in the age of molecular virology, instead of traditional infectious virus titration, the viremia level is far more frequently determined by measuring the amount of vRNA copies detectable. Because not all full-length viral genome copies are infectious, ideally such an assay is validated by running a parallel infectious assay (i.e., plaque assay) for a fraction of samples with an infectious reference, such as YFV (17D). In one of those reports, the viral load was measured on cadavers of NHPs which had died of YFV infection during the 2017–2018 outbreak in Brazil [114]. The three problems in this report are (i) that no parallel infectious titer assay with a reference virus was performed, (ii) the period between death and discovery of the cadaver was unknown, and (iii) decay of vRNA was not taken into consideration. In another article, it was reported that vRNA was detected in urine up to day 24 and viremia lasted for 20 days [115]. 

### 4.5. The Vertebrate Reservoir and Virus Persistence

Arbovirology today is based on the arbovirus concept and guidelines established by the WHO [1]. However, as mentioned earlier, the WHO definition of arbovirus largely reflects Meyer’s zoonotic principles. One task that has never been completed was confirmation of the existence of the vertebrate reservoir for RNA arboviruses, including YFV. Meyer himself continued to look for the reservoirs in vain [116]. His struggle is well illustrated in one of Meyer’s publications on zoonoses caused by arboviruses. While at first listing primates, marsupials and ungulates as principal reservoirs in jungle YF in one table, in subsequent passages, he identified *Haemagogus* mosquitoes as the main reservoir. Later in the article, he came to conclude that because “monkeys die or become immune, their participation in the cycle is fleeting and that jungle monkeys are intermediary hosts” [61].

Nevertheless, the supporters of the vertebrate reservoir for YFV and other arboviruses increased confidence further when the WHO fully adopted Meyer’s principles in the arbovirus definition [1]. However, because a vertebrate chronically infected by an arbovirus had never been found in the field, for a pure curiosity, some even attempted unusual studies, just to find out if continual BT is possible at least under laboratory conditions. In one such study, a *Haemagogus* species and a species of squirrel monkey in South America were used to reproduce a continuous cycle of BT of YFV. A year-long continual transmission experiment was planned, but was terminated only after five cycles, because monkeys had to be replaced continuously, as they kept dying or becoming useless because of the development of immunity [117,118]. 

However, looking back, as early as 1920, Carter declared that “the evidence for the existence of a permanent reservoir host seems to me to have no basis in observations” and “the evidence against it is considerable” [63]. A number of workers who spent many years on YF similarly reached the same conclusion [22,60,119]. John D. Gillett called the notion of YF reservoirs a “monkey myth” [120]. Thomas P. Monath stated that “whereas NHPs in Africa play an essential role in the amplification of virus transmission, there is no evidence that latent infections contribute to recrudescent virus activity in nature; and monkeys, therefore, do not constitute a true virus reservoir” [68]. Thus, it is remarkable that the tradition of referencing NHPs as reservoirs of YFV and other vertebrates for all other arboviruses continues undiminished today.

### 4.6. Direct Transmission (DT) of YFV in Vertebrates

Because DT has been exclusively investigated in vectors, little attention has been paid to the old studies on the possibility of DT in vertebrates. Aerosol transmission was well recognized because of many accidental infections among laboratory workers [121,122]. These publications were later used to establish a safety standard for laboratory workers. The interest in oral YFV infection in wildlife arose because some monkey species (i.e., baboons) and predatory birds (i.e., crowned hawk eagle) in Africa engage in cannibalizing monkeys. A study confirmed oral infection of YFV in NHPs but not in many other groups of vertebrates they investigated [123]. 

### 4.7. Viral Dispersal by Humans

In the early part of research history, the two major means involved in the transoceanic dispersal of YF were identified as by ship and by aircraft. Later, the roles of automobiles and trains in transcontinental dispersal were studied but found inconclusive because of the lack of information available [74]. While ship and aircraft have been traditionally identified as the principal means of the global dispersal of tropical infectious agents, Patrick Manson was one of the pioneers who conceived the idea that travel by infected humans itself is basically responsible for the dispersal of tropical diseases. Thus, he was well ahead of time, as dengue, chikungunya, and Zika fever became the major global concerns in modern times much later. He arrived at this concept through many years of examining at the Seamen’s Hospital in London hundreds of seafarers returning sick from the tropics. 

#### 4.7.1. Ship

The dispersal of YFV and *Ae. aegypti* from West Africa to the New World in association with the slave trade has been a popular topic in the historical account of YF dispersal [71]. What has not been well known is the identity of a man who advocated abolition of slavery not only on moral grounds but on importation of YF. This man was Mathieu F. M. Audouard [124]. He was a French military physician who spent much time in Rio de Janeiro. He campaigned hard to stop the slave trade because of its strong linkage to YF outbreaks in the Americas [125]. 

Shipboard YF cases started to decline after steamships began to replace sailing ships. After 1893, sailing ships virtually disappeared and so did YF outbreaks from many seaports in North America, with only one exceptional outbreak in New Orleans in 1905. The last outbreak in Barcelona occurred in 1870; and the outbreak in Trieste (then part of the Austro-Hungarian Empire but now part of Italy) in 1894 marked the end of YF in Europe [126]. Cuba was free of YF in 1909.

When the U.S. announced a plan to construct the Panama Canal, the plan alarmed Manson. He addressed a dire warning about the consequence of YF spread to tropical Asian cities (infested by *Ae. aegypti*), where virtually nobody had immunity [127]. In response, the U.S. decided to establish a strict quarantine system at the entrance to the Canal for inspection of ships and quarantine of suspected infected passengers. Manson further suggested deforesting and thoroughly draining a ½ mile wide zone on both sides of the Canal. Carter discounted the Manson’s concern and felt that an effective quarantine at the Canal would be sufficient. Furthermore, several countries in the Americas ratified an international health convention to prevent the dispersal of YF. Nonetheless, one case of the autochthonous transmission of YF occurred in Hawaii in 1911. The vessel involved did not pass through the Panama Canal but came from Peru with a stopover in a Pacific port in Mexico [128]. 

#### 4.7.2. Aircraft

As air travel began to replace ship travel, inspection of the aircraft from the YF zone was implemented. Establishment of an “antiamaryl aerodrome” (“antiamaryl” meaning anti-YF) was one of the strategies adopted in Africa [73]. In the U.S., aircraft inspection by the quarantine service began in 1931. The dispersal of *Ae. aegypti* by aircraft became more intense after WWII. In French Polynesia, many islands initially free of this mosquito were quickly infested by it after an inter-island air service was inaugurated [129]; dengue outbreaks quickly followed; and DENV established a long persistence in some islands. Unlike dengue, however, increased air transportation did not expand the YF zone. One possible reason behind the difference between sailing ships and aircraft was that while sailing ships provided a “miniature urban environment” (Section 8.2. (iii)) onboard for YFV transmission to proceed during the long journey (often lasting more than 2 months with stopovers), aircraft did not provide such an onboard “ miniature urban environment”.

### 4.8. Acquired Immunity and Social Distancing

In the Mediterranean locations in Europe where YF outbreaks occurred multiple times, a consensus began to emerge that immunity to YF is life long. This was promoted by a physician in Gibraltar after the 1804 outbreak. He advised “exposing children to YF in the hope of minimizing their risk of dying from it”. In the time of the outbreak, the residents who survived the YF attack were given a “fever pass” to move around anywhere and work normally, while many of the residents who were not immune were confined to living in encampments where movement and their work-related activities were severely restricted [130]. When YF outbreaks returned in 1828, death rates per 1000 for the residents in the encampment and those who chose to live in their homes and move freely were 19.75 and 103.90, respectively [131]. This proved that social distancing reduces YF cases considerably. It is interesting to note that, following this old tradition, during the COVID-19 pandemic that began in 2020, all adult residents of Gibraltar were immunized with a mRNA-based vaccine and issued a “COVID-19 passport” valid for unlimited travel to UK and in the European Union [131]. In the current pandemic of COVID-19 in South America, most governments enforced social distancing at one time or another. It is of interest to note that among the three viruses (CHIKV, DENV, and ZIKV) transmitted by *Ae. aegypti* concurrently, the sharpest decline in incidence rate was observed in DENV [132]. 

## 5. Environments

The WHO’s arbovirus concept and guidelines for research [1] emphasized the importance of the environment as an inseparable element in BT. However, it did not elaborate much. The critical role of the environment in terms of its importance in a variety of topics ranging from disease dynamics in epidemiology to virus evolution was actually more clearly recognized by others [50,133,134,135].

### 5.1. Classification of the Environment Based on the Combination of the Ecosystem, the Landscape, and the Pattern of Human Habitation

Historically, as far as YF transmission is concerned, it was George M. Findlay who was the first to clearly identify three types of environments: urban, rural epidemic, and rural endemic [74]. Others used similar but more specific classifications. Strictly speaking, whichever classification system is used for environments, ambiguity cannot be completely eliminated because of considerable overlapping. Nonetheless, Findlay’s classification is adopted in this review, with only a minor modification (by changing “rural endemic” to “sylvan”) (Figure 1). To avoid unnecessary confusion, in this review, the following geographic–epidemiologic terms are used. 

### 5.2. Terms Used 

The “YF zone:” This refers to the vast Sub-Saharan belt across the Tropics of Cancer and Capricorn in Africa and from the Darién Gap of Panamá to Bolivia on the eastern parts of the Andes in South America. Central America (north of the Panama Canal) and the Caribbean islands where YF outbreaks occurred in the past are not included. The zone defined by the WHO includes the areas at risk located around the periphery of the zone, which change slightly from time to time. The boundary of the zone follows the earlier guideline set by a panel of advisors [136].

“Enzootic/epizootic focus”: This refers to a particular geographic area within the YF zone where enzootic transmission (or persistent maintenance) of YFV occurs. Epizootic transmission evolves from an enzootic focus. In South America, enzootic foci are heavily localized along the eastern foothills of the Andes. The locations of epizootic transmission among howler monkeys are sometimes roughly identifiable because howling of susceptible monkeys stops due to high fatality and forest becomes silent. However, because it is nearly impossible to segregate epizootic locations from enzootic locations; both are jointly described as “enzootic/epizootic”, except when segregation of the two is more appropriate. Furthermore, in this review, it is assumed that in each YF zone, there exist many enzootic/epizootic foci which not only dynamically change the location and transmission activity over time but undergo merger with other focus (or foci) or spontaneously proceed to localized extinction. This dynamic pattern has been characterized as “wandering” or compared to the protrusion/retraction of pseudopod of amoeba [22].

The existence of multiple epizootic foci was deduced from the lack of synchrony among multiple outbreaks that occurred in a given period in a particular region of Africa [68]. A small number of the enzootic/epizootic foci, however, have been stationary for a long period. Two good examples are the forests in the Andes in Peru and the southeastern part of Sénégal. The use of a popular expression, “endemic”, as in “endemic area”, is avoided as much as possible because of ambiguity and frequent, unscientific generalization, except when the probability of the existence of the virus is reasonably certain as evidenced by at least sporadic human or NHP cases or seropositivity in the past in the areas mentioned. In fact, “endemic area” is not even used in International Health Regulations. 

### 5.3. Sylvan Environments (Figure 1)

Ever since Fred L. Soper coined the term “jungle yellow fever” for the type of transmission without *Ae. aegypti* in 1934 [137], the word has been popularly used for sylvan transmission. Taylor objected to the use of “jungle” because he felt that ecologically “jungle” referred to impenetrably thick vegetation (and hence little penetration of sun light) that grows typically around the periphery of tropical rain forest. There, the interactions between insects and NHPs do not occur. Actually, arbovirus transmission occurs in the interior of the rainforest, where the canopy is taller and more sun light penetrates [22]. While Taylor was partially correct ecologically by relying on the old Hindi meaning (“an uninhabitable place”), conventional usage by the general public at his time most often meant just “tropical rain forest”. Findlay called YF transmission there “rural endemicity” [74]. While in many sylvan environments in South America, YF transmission activity is seasonal and depends on the arrival of rainy seasons, in parts of Colombia and around the Amazon basin, transmission is year round, as shown in an investigation of American Indian tribes in the Amazon. A similar year-round environment is also found in the rainforests in Africa.

Despite the importance of sylvan environments as the place where YFV is maintained, the amounts of information necessary to elucidate the mechanisms of viral transmission and of perpetuation are too scarce. Thus, exactly how epizootic outbreaks and its spillover occur are poorly understood. As an example, breaking down serologic survey results of NHPs into age group to deduce timeline of the past exposure to YFV is not possible because one cannot tell accurately their age, except the infants. Determination of the population sizes of most species of NHPs is also difficult because of their constant movement.

#### 5.3.1. Sylvan Environments in the Americas

During the expedition to South America in the early part of the 19th century, Alexander von Humboldt observed YF outbreaks in coastal areas of Venezuela and Vera Cruz, Mexico. As a broad naturalist, he was interested in YF and documented an old outbreak (1544–1545) in Mexico and investigated the pathology of YF [138]. Though he had no knowledge about the YF transmission mechanism, it is an interesting coincidence that he collected the type specimen of *Sabethes chloropterus* (a YF vector) in 1819 in Ecuador and type specimens of several NHPs elsewhere including marmosets and night monkeys which are now known to be susceptible to YFV. 

Drawing the exact geographic boundary of the YF zone in South America is difficult and is a subject for debate. In Brazil alone, it is applied by some to nearly two-thirds of the territory, covering tropical forests, Amazon Basin, Pantanal (tropical wetland), more arid Cerrado, degraded fragments of the Atlantic Forest, and riparian forests (equivalent to gallery forests in Africa). In their definition, the only area not in the YF zone is the interior of the northeastern states where the dry landscapes are locally called “sertão” and “caantinga”. In Peru, the zone covers eastern parts of the Andes as high as 2300 m above sea level. The dynamic development of epizootic foci from enzootic areas resembling protrusion and retraction of the pseudopod of amoeba was most frequently observed in Brazil [139]. 

Although little is known about exactly when the sylvan transmission of YFV was established in the Americas, the oldest known outbreaks occurred in Santa Cruz, Bolivia in 1887, in Muzo, Colombia in 1885 or 1907, and in Socorro, Colombia in 1929 [140,141,142]. In Brazil, although the 1932 case in Espírito Santo State is remembered among most researchers [143], apparently Adolfo Lutz recorded YF transmission without *Ae. aegypti* in Campinas, Sao Paulo State in 1889 [144]. 

Vectors: The mosquitoes belonging to the genus *Haemagogus* are the dominant vectors, while members of the genus *Sabethes* are also important [145]. The two genera of mosquitoes are phylogenetically very close to *Aedes* mosquitoes, based on mitochondrial DNA; and their geographic distribution is restricted to the Americas. It was Allen W. Burke of the RF who discovered *Haemagogus spegazzinii* as a sylvan vector in Brazil for the first time [14]. The diurnal feeding behavior of many *Haemagogus* species corresponds to the daytime activity of the hosts. More recently, however, it was discovered that the peak biting activity of *Sabethes chloropterus* in parts of Brazil exceptionally falls between 4:30 and 5:30 p.m., suggesting local variation in diurnal feeding patterns.

The long flight range (11.5 km per release) of *Haemagogus janthinomys* [146] is known to be contributing to a rapid spread of YF over a long distance in South America. Many canopy-dwelling mosquitoes also descend to the ground to bite animals and humans. Those arboreal vector populations are multivoltine, each brood hatching at a different time even with a small amount of rain. This is considered an example of evolutionary adaptation to survive long dry seasons by means of embryonic diapause. *Sabethes chloropterus* as a YFV reservoir has been supported by repeated isolation of the virus from sylvan mosquitoes during cold and dry seasons in Brazil when transmission activity was absent. In a recent report, during a peak of dry seasons in semi-arid Cerrado (in the State of Minas Gerais), YFV RNA could be persistently detected in *Sabethes chloropterus* mosquitoes [69]. Although persistent detection of viral RNA alone is not a definitive proof of viral persistence, this is still a very intriguing report. 

Hosts: Most NHPs are susceptible to YFV infection, which results in high rates of mortality in many species including howler monkeys. Those neotropical monkeys occur exclusively in the Americas. As mortalities of many NHPs in the Americas are high, it takes many years (approximately 5–12 years but highly unpredictable) for the collective population of all hosts to recover to a level sufficient to initiate another epizootic. In Trinidad, where the total population size of all NHPs is much smaller than those in South America, between 1913 and 1990, large outbreak occurred at intervals of 19–23 years, while small outbreaks occurred at intervals of 3–10 years, respectively [147]. This is contrasted to the NHPs in Africa, where it takes only 3–5 years because of very low mortality. The minimal size of the population of any NHP is generally unknown; but in Trinidad, where outbreak occurred multiple times but not consistently, the total population size of NHPs was estimated at one time to be approximately 2000–3000 [148]. It is stressed, however, that the concept of an interval for the host population to recover in the Americas should only be applied to major outbreaks but not to small outbreaks (typically less than 30 cases). This is proposed because the significance and frequency of epizootics, as well as the length of the interval for the recovery of the host population, varied among authors considerably because of the lack of definition.

Generally, research on those NHPs has been hampered by the difficulty of breeding them in captivity. Spider monkey and marmosets (*Callithrix* spp.) are exceptionally amenable for laboratory experiments, but many species of the latter primates often do not live long in captivity. The interest in those lower primates as possible reservoirs arose because of higher reproductive rates (a qualification as a reservoir in old times). They are generally susceptible, and some inhabit urban areas. Hybrid species are also kept as pets. Thus, Davis called attention to their possible role in urban YF in Brazil in 1930 [149]. The same concern was recently raised again in the wake of the 2016–2019 outbreak because of high seropositivity [114]. However, a negative comment was also raised because of less virus sensitivity [150]. 

Monkeys move in bands. The size of the band varies, but 10–30 is common, though it may be as small as 6. *Saimiri* monkeys are known to live in bands of 20–30. As for foraging range, compared with some NHPs in Africa, it is generally much shorter than 3 km.

The number of human cases in this environment is usually small. Those who enter forests for work (i.e., logging, mining, oil drilling, highway construction, etc.) and for eco-tourism may be infected. However, indigenous American Indians are exceptional permanent residents in this environment. In a serologic study of these people in the Brazilian Amazon, 56–85% of them over 15 years were YF positive at a rate of 3% increase per annum [151]. In the 1912 outbreak in the Department of Chuquisaca, Bolivia, mortality among the American Indians was very high, nearly decimating them [22].

Virus: Two genotypes exist in the Americas. A study revealed that while in situ evolution (sign of enzootic perpetuation) occurred multiple times in Trinidad, evidence of the repeated introduction of the virus from South America was confirmed [152]. In the Peruvian Andes, the lineage has largely remained isolated without much genetic inflow from other regions, but occasional outflow of the virus apparently occurred. 

Spillback: How YFV (well adapted to urban transmission in West Africa), after dispersal to South America, reversed back to sylvan transmission remains an unresolved but very important question [153]. Three hypotheses have been proposed. In the first proposal, the reversal was the result of viral adaptation to intermediate (bridge) vectors that inhabit between forest and urban environments [154]. This was disputed by others [155]. In the second hypothesis [156], direct adaptation of the virus to sylvan vectors occurred in the enzootic/epizootic foci. However, exactly how it occurred was not elaborated. According to the third hypothesis [73], in the first stage, during YF transmission within the urban environment, contact of sylvan vectors with YFV transmitted by *Ae. aegypti* occurred because of the close proximity of sylvan environments to urban communities along the coastal areas at the time. The two key features of this hypothesis are a very long flight range and indoor feeding of some sylvan vectors. In the second stage, occasional (but still incomplete) adaptation of the virus to sylvan vectors occurred. In the third stage, full adaptation of the virus to sylvan vectors occurred in the urban environment. As urban areas expanded, the distance between sylvan and urban environments grew longer beyond the flight range of sylvan vectors. Then, the populations of sylvan vectors infected with the virus were separated from the urban environment. Clearly, more research to obtain supporting evidence is needed for all hypotheses. 

In some islands in the Caribbean (excepting Trinidad), colonies of African monkeys are found. These NHPs (i.e., green monkeys) were originally brought there by the slaves as pets. However, because no sylvan YF vectors occur there, spillback was not possible.

#### 5.3.2. Sylvan Environments in Africa

The sylvan transmission of YF was discovered in Bwamba County in Uganda, followed by the discovery in equatorial forests of Africa between 1930 and 1949 [157]. Typical rainforest and gallery forest (the fringe of forest along the river banks) are the major landscapes involved.

Epizootic focus is characterized by the rapid increase in the size of the vector population which coincides with the arrival of wet season. Epizootic/epizootic foci dynamically shift in geographic location, as sizes of vector and host populations change, resulting in “wandering” pattern [22]. In the equatorial rainforests which extend from Guinea to Uganda and as far south as Northern Angola, year-round transmission occurs between *Ae. africanus* and NHPs. 

Vectors: *Ae. africanus* is a major vector in Africa. As this species is a canopy breeder like *Haemagogus* mosquitoes in the Americas, they descend to the ground and bite humans. Their crepuscular feeding activity is well coordinated with resting time of the arboreal hosts (NHPs), another example of evolutionary adaptation [158]. Other primary vectors include *Ae. luteocephalus*, *Ae. taylori*, *Ae. opok*, and *Ae. furcifer.* As described earlier, *Ae. aegypti formosus* in the forest is zoophilic and has little to do with human infection. All primary vectors occur exclusively in the African continent. Among the secondary vectors, such as *Ae. bromeliae* and *Ae. vittatus*, the former occurs also in Madagascar and Comoros, while the latter occurs additionally in tropical Asia from Pakistan to China.

Hosts: Among African NHPs, Barbary macaque (*Macaca sylvanus*) in northern Africa and Gibraltar is exceptionally susceptible to YFV, as this species is taxonomically close to rhesus monkey in Asia [159]. In 1828, YF outbreaks in Gibraltar and unusual deaths of pet monkeys including lemurids were recorded [160]. Edward Hindle also learned that many Barbary monkeys died during that epidemic [161]. However, confirmation of this history is difficult because apparently no scientifically verifiable document was left in Gibraltar.

Shortly after the isolation of French strain of YFV, Auguste Pettit and Georges Stefanopoulo of the Institut Pasteur determined that most other NHPs in Africa support YFV replication, but their mortalities are low, although death may occasionally occur in laboratory experiments. Most monkeys present viremia lasting for a few days but without icterus or blood vomit. Their low mortality, in contrast to higher mortality in the NHPs in the Americas, has been attributed to adaptation to YFV over a long period of co-evolution. *Colobus* monkeys demonstrate a prolonged viremia (up to 9 days) and are probably involved in YF transmission in the forest canopy where *Ae. africanus* is the major vector. Mangabeys and guenons are probably involved, too. Some baboons were once suspected to be hosts of YFV. *Patas* monkeys are found in dry savanna, gallery forest and savanna woodland. They travel long distances in bands of 5–30 members for foraging across savanna as far as 16 km/day and covering an area as large as 300 km^2^ [162]. Lemuroids (galagos or “bush babies”), which occur broadly across Africa, generally support YFV replication. Though once they were strongly suspected to be the hosts of YFV, subsequent research did not support the idea. 

In parts of Africa, functional differentiation among NHPs was observed with respect to YFV transmission. Lowland *Colobus* monkeys are involved in transmission in the interior of the forest, while red tail monkeys play a role of bridge vertebrate to humans. Generally, exclusively arboreal monkeys revealed a higher level of immunity than terrestrial monkeys (such as grivet monkeys). 

The minimal population size of NHPs (either by individual species or by the sum of all species) that supports viral persistence in enzootic/epizootic focus has been one of the research interests. Earlier, it was proposed that for a stable, focalized sylvan transmission of YFV to occur in a community of 130 monkeys, the minimal birth rate of 400 per 1000 would be necessary [163]. However, the low fertility rate of most species of NHPs is a limitation to satisfy such a hypothesis. Though reliable data are scarce, in an old study at Semliki Forest in Uganda, the total population size of all NHPs combined was estimated to be approximately 400/square mile. In a study in Uganda, the seropositivity of NHPs to YFV was almost 80%, and yet their total population in a group of forests combined was only 20–50 [157].

In the gallery forest of Bozo, Central African Republic, epizootic outbreaks occurred at intervals of less than 10 months (1974, 1977, and 1978). This result was found questionable, because there were less than 10 monkeys in the gallery forest. In Kédougou, Sénégal, YF outbreaks occurred between 1977 and 1978. After the 1977 outbreak, 48% of NHPs were still susceptible. Only when herd immunity level reached 95% in 1979 did epizootic outbreaks finally stop [162]. Spillover from epizootic outbreaks has occurred in the transitional environments at less than 5-year intervals (Section 5.4.2). 

The combination of a generally high seroprevalence in NHPs, the small size of the NHP population in a given environment, and the low fertility of most NHPs was a major puzzle for understanding the mechanism of YFV persistence in enzootic foci. Two possibilities were proposed to explain viral persistence. One was the possibility of a second type of sylvan transmission cycle involving hosts other than NHPs. No evidence to support this hypothesis was found. The other possibility was the periodic reintroduction of YFV by wind-borne mosquitoes [157,164]. Clearly, more research is needed.

### 5.4. Transitional Environments (Figure 1)

In environments neither characterized as strictly sylvan nor as urban, YF transmission occurs in both Africa and the Americas. In those environments, some of the vectors and hosts of YFV may not be shared with either one. YF transmission in these intermediate environments was more clearly recognized by George W.M. Findlay as “rural epidemicity”, because of the inconsistent occurrence of YF [74]. 

Multiple terms have been used to refer to YF transmission in these environments characterized by the intermediate ranges of the landscape, fauna, and vegetation in relation to the sylvan or urban environment. They have sometimes been called “rural”, “feral”, “zone of interface” [22], and “zone of emergence” [165]. The characteristics represent savanna, forest savanna, riverine (gallery) forest, rural or feral landscapes, and “open” environments (which include cleared land for agriculture or industrial development) next to forest or savanna and arid environment with much less vegetation. Because subdividing these environments creates more ambiguity due to overlap, in this review, these diverse environments are lumped together as “transitional environments” which was coined earlier [68]. These are the environments where spillover epizootic outbreaks and/or epidemics of YF occur at irregular intervals. 

#### 5.4.1. Transitional Environments in the Americas

In the 1930s, YF outbreak sites were classified as either “rural” or “urban”. Some of the “rural” outbreaks were subsequently interpreted as “sylvan”. However, other old rural outbreaks were re-classifiable to true transitional environments [137,166,167]. A “rural environment” is typically characterized by clusters of a small number of houses separated with a variable distance in an irregular pattern. Exposure to wildlife was a shared feature. In Brazil, arid environments (such as pampas in the south and savanna-like Cerrado), degraded patches of the Atlantic Forest, and more humid environments (such as Pantanal and lower Amazon Basins) fall in this category. Northeastern parts of Colombia, the Misiones Province of Argentina and the eastern plain of Bolivia are other locations of frequent YF activity. The size of YF outbreaks varies, but is sometimes considerable. In the 1929 outbreak in Socorro, Colombia, 34 deaths and 150 symptomatic cases were registered. However, a closer retrospective examination revealed an actual size of approximately 2000 symptomatic cases [141,151]. The 2016–2018 outbreak in Brazil was the largest in more than several decades in terms of the prevalence (2170) and the number of deaths (932) [155]. In contrast, the magnitudes of the outbreaks (1997–2008) in such countries as Bolivia and Paraguay were small (27 cases/13 deaths and 25 cases/6 deaths, respectively).

The “YF corridor”: More recently, for the occurrence of YF outbreaks between the transitional environment and the urban environment, the term “YF corridor” was coined [73]. The word refers to an imaginary combination of opportunities unknowingly generated by human activities, the landscape or other geographic/economic specifics (in both the Americas and Africa) to bring YF closer to the urban environment. Human activities include migration into urban areas by traders (of agricultural or forest commodities), seasonal workers, nomadic people, and others from transitional environments to the urban environment. Other human activities are modifications of sylvan environments to facilitate contact between vectors and humans, such as deforestation, lumber business, mining, oil drilling, dam or highway construction, and industrial or agricultural development along the fringe of the forest. Also included are all roads and transportation systems linking sylvan environments with urban areas that are conducive to the introduction of YFV. This imaginary corridor appears before an urban outbreak and disappears at the end of outbreak.

In the southeast region of Brazil, due to rapid urbanization, the distance of the YF corridor was shortened, which likely contributed to increased YF cases [168]. The closeness is illustrated by the fact that in some large cities, residents living in higher buildings occasionally observe monkeys frolicking in the canopy of the forest nearby. Theoretically, the shorter the distance of the corridor (i.e., <2 km), the higher the probability of the emergence of a YF case [169]. In one study, the mean distance between the locations of positive NHP samples and human cases was 11 km. Interestingly, this corresponded well with the flight distance of *Hg. janthinomys* [146,170].

In the Americas, the interval between spillovers is unpredictable but is generally estimated to be 7–12 years, which reflects long periods required for the susceptible NHP populations to recover due to high rates of mortality. One major feature of spillover there is long-distance dispersal. Long-distance flight of *Haemagogus* vectors is one of the factors contributing to this unusual pattern of dispersal, though human activities and perhaps wind may also play a role. Historically, in the 1934–1940 outbreak in South America, the spillover travelled at an estimated rate of 200 km per month at its peak, for a total distance of 1500 km [22]. In the outbreak in Central America (1948–1957), foci travelled 1550 miles from Panamá to the Guatemala-Mexico border at approximately 0.5 mile/day. According to Pedro Galindo, apparently this long outbreak was caused by multiple waves [171], suggesting a chain of a few consecutive outbreaks each from a different epizootic focus rather than a single and long outbreak. 

More recently (2016–2019), history repeated itself in Brazil. The spillover spread at 3 km/day for a distance of 600 km. In a study conducted in the State of Minas Gerais, the travel distance was 1.42 km/day (range 0.1–6.9 km); and the critical distance between house and forest for a case to occur was 100 m [172]. The host-finding behavior of *Haemagogus* vectors also contributes to transmission. As an example, in the area near Santa Cruz, Bolivia, these vectors were found to engage in indoor biting, just like *Ae. aegypti*. Moreover, they breed in barrels and water tanks. Similar observations were made in Brazil and Panamá [73].

Though the rapidity and long-distance spread of spillover in the Americas are remarkable, YFV involved in outbreaks does not remain long in any fixed location. The maximum period of persistence in the transitional environments in Brazil during the 2016–2019 outbreak was 3 years [173].

#### 5.4.2. Transitional Environments in Africa

These environments are found in the Sub-Saharan belt, stretching from West Africa to Ethiopia, Sudan and Kenya. In West Africa, YF spillover occurs in or near human settlements in savanna, fragmented (degraded) forest, forest-savanna mosaic and dry savanna. Cross-border migration of people is a significant factor in YF epidemiology. The migrants are traders, religious pilgrims, seasonal migrants during specific celebration, refugees of political or economic instability, and nomadic people in such countries as Nigeria, Burkina Faso, Ghana, Sénégal and Sudan. Although the early history of YF in the rural areas of Sudan is poorly understood, historical records strongly suggest that YF was prevalent in Kordofan of Sudan by mid-19th century [174]. Nearly all outbreaks in Sudan have occurred in transitional environments, the largest being the 1940 outbreak in the Nuba Mountains involving nearly 40,000 cases and approximately 1500 deaths [174].

The mechanism of the emergence of spillover in the “zone of emergence” was well documented by Max Germain and his French colleagues [165,175]. Another example was the discovery of the intermediary role of *Ae. simplex* complex mosquitoes (principally *Ae. bromeliae*) in Uganda [41]. However, elsewhere in East Africa, exactly how YF outbreaks occur still remains unresolved despite increased epidemic activity in Kenya lately. Although YF does not occur in highlands, multiple tribes living apart at a high altitude in western Uganda were found to have a high seropositivity. These tribes descend to lowlands periodically to sell their commodities at a market. This puzzle over the high rate of YF exposure of the highland tribes was solved when investigators found out that when the tribes descended the mountain, all tribes had to pass through a shared road along gallery forest and banana plantations to reach the market and that the plantations as a whole was an epizootic focus [157].

In Africa, with respect to transitional environments, what may appear to be a common observation in certain regions of a country may not necessarily be observable in other regions in the same country or in the comparable environments in other countries. As an example, while *Ae. aegypti* was found at one time to be the major vector in some rural epidemics in Sénégal, it was a minor vector in other epidemics in transitional environments elsewhere in West Africa. Thus, the relative importance among four vectors (*Ae. africanus*, *Ae. luteocephalus*, *Ae. furcifer* and *Ae. aegypti*) varies from region to region [176]. Still, *Ae*, *africanus* is most important in these environments, as its flight distance is nearly 3 km in 4–8 days. *Ae. bromeliae* and *Ae. furcifer* are also important because of their role as bridge vectors. In West Africa, epizootic outbreaks have occurred at shorter intervals (less than 5 years) of YF inactivity than in South America. Small outbreaks occurred once every few years in Nigeria or every 2–5 years (range 1–17 years) in Ghana. In Kenya, the interval is much longer, and the next outbreak is very much unpredictable.

### 5.5. The Urban Environment (Figure 1)

Before comparing the current status of urban transmission between the Americas and Africa, it is worthwhile to review briefly how the magnitude and pattern of urban transmission as well as YF control changed over a few centuries. This history is useful for evaluating the possible return of YF to urban areas in Europe and North America.

#### 5.5.1. A Brief Review of the Urban Outbreaks in Europe, North America and South America before 1910

In the 18th and 19th centuries, urban outbreaks in temperate areas occurred in seaports temporarily infested by *Ae. aeygpti* in warm seasons. Since the late 18th century, the pattern of YF spread in some seaports was investigated block by block, which established a correlation of higher incidence with residence closer to the wharves and arrival of sailing ships from the tropics. This was observed by Samuel Jackson in Philadelphia during the 1793 outbreak [177], in New York in 1795 by Valentine Seaman [178], and in Barcelona in 1870 [179]. Thus, geographic medicine evolved in YF investigation much earlier [180] than the well-known source (shared well used for drinking) identification of cholera agent by John Snow in England in 1854 [181]. More than a few accounts of YF spread by travelers within cities in Spain and from Spain to Gibraltar suggest a secondary spread by human movement, provided that all communities involved were infested by the vector. This was also observed multiple times in the southern regions of the U.S., including suspected spread by trains or ships serving major rivers.

All epidemiologic statistics well before the definitive establishment of YF etiology, the development of diagnostic methods and the application of case definition were crude estimates.

Because nearly 152 colloquial names were used in various parts of the world, it is unknown exactly how many of the outbreaks retrospectively identified were true YF. Some of the accounts of YF featured in many books or other types of documents were found to be equivocal [3].

Nevertheless, the outbreaks characterized by the development of jaundice and/or blood vomit, such as “yellow fever” coined in 1750 by Griffith Hughes in Barbados [182] and “vómito negro” used in Cartagena, Colombia beginning in approximately 1729 [183], were probably more accurate etiologically. Although “vómito negro” was known among physicians in some Hispanic countries and even in Brazil, it is likely that in many other areas affected by YF, most physicians probably never heard of such names because of limited means of international medical communication.

Regarding the size of outbreak, three factors were the common variables, the size of the population, the proportion of the residents (mostly affluent) who fled for safety, and the quality of demographic registry. All three variables being more or less equal, larger numbers of incriminating sailing ships that arrived increased prevalence and mortality. If multiple incriminating ships arrived each at a different time, sometimes bimodal waves of YF attack that prolonged outbreak much longer were observed [179]. In the cases of consecutive outbreaks for several years in row or longer, such as in Gibraltar (1800–1804, 1810–1814, 1817–1826), the yearly introduction of the unidentified agent was retrospectively suspected [184].

Before the mid-19th century, in most affected countries, the prevalence and the number of deaths were based on the parochial registries without mortality differentiation. In the outbreak in Buenos Aires in 1871, the death toll was approximately 14,000 [185]. According to a historian, a death toll of 5000 in the southern parts of the U.S. actually could have represented anywhere from 20,000 to 30,000, because of the difficulty of identifying YF patients [186]. The mortality figure in the outbreak in Barcelona in 1821 ranged from 3251 to 8000. In the outbreak of 1824, while the earliest record was 17,000 deaths [124], the estimated figures in more recent publications ranged from 6000 [187] to 20,000 [188]. As for CFR (in %), in Gibraltar, where very detailed records of epidemics were kept, the figure in the intense epidemic period (1804–1828) was approximately 17 [184]. In southern states of the U.S., it was, on average, approximately 20 [186].

In Europe, from the early part of the 19th century, many countries began to enact a demographic registry law to take over all personal or family matters including health and mortality, although ecclesiastic registration of mortality record was exceptionally established earlier in 1532 in London [189]. The severity of the 1793 outbreak in Philadelphia prompted the City of New York to establish the first municipal Board of Health Department. In the wake of the 1795 outbreak there, the Board erected a hospital to exclusively treat YF patients in an estate known as Belle Vue and named it “Bellevue Hospital” (where William C. Gorgas would train many years later). The lazaretto quarantine hospital (the oldest in the U.S.) built in 1801 at the entrance to the port of Philadelphia was similarly used for YF patients. Baltimore and Boston followed this movement quickly. New Orleans established the first State Board of Health in 1855 after six earlier municipal Boards of Health failed to function. In Gibraltar, a similar board of health existed as early as 1805 [184].

The major reasons for the decline in outbreak in seaports in Europe and North America were the end of the geopolitics (colonial wars) among European countries [186,190], the decline in trade (in particular sugar) with the Caribbean and South America, replacement of sailing ships with steamers [49], enforcement of more rigorous quarantine practices, and the establishment (from late 19th to early 20th century) of international health convention among European countries and separately among some American states. After 1945, implementation of the International Health Regulations and vaccination programs probably helped prevent its recurrence. Today, though large-scale urban outbreaks occur infrequently, the magnitude of annual YF statistics released by the WHO is, ironically, still largely based on multiple assumptions, partially resembling the state of affairs in the 19th century, not because of the lack of science but because of under-reporting.

#### 5.5.2. The Problems and Importance of the Definition of Urban Area

Reporting “urban YF” has been a source of controversy for a few reasons. First, the definition of the “urban environment” varied among authors and countries. Thus, the communities called “urban” in one country in Africa by some were perceived by others as “rural communities” or “a group of villages”. In Africa, migration of a considerable number of people further complicates the definition of “urban area” based on population size. To make the matter further complicated, in some outbreaks in African “rural environments”, unlike in the Americas, *Ae. aegypti* is involved. Furthermore, as the human population grew, the number of more specialized terms (such as urban, peri-urban, and suburban) began increasing. In modern times, the definitions adopted in some countries (such as Brazil for administrative reason) based on the size of the population, density of household per given space, distance separating clusters of houses, and others were available. However, such a definition has been rarely used in epidemiologic reporting of YF in the past. This is a shared problem of all diseases vectored by *Ae. aegypti* including dengue, chikungunya, and Zika fever (Section 6.6).

Despite the problems, still it is important to define “urban areas” at least for practically identifying a pattern of YF transmission. One pragmatic way to designate the urban transmission of YF is the application of the following criteria: (i) clinical confirmation of patients based on more reliable symptoms and preferably also on one or more reliable diagnostic tests; (ii) ample infestation of the community by *Ae. aegypti*; (iii) lack of travel history to YF enzootic/epizootic focus by the majority of patients at least for 3 weeks before the onset of illness; and (iv) evidence of secondary transmission by *Ae. aegypti* within the community after the detection of an index case. In this review, reports of urban transmission are accepted if these four criteria were generally (even if not completely) met. It would be most ideal if the traditional medical detective strategy through back tracking investigation of the chain of human contact involving YF patients [105] was also performed for an absolute confirmation. This set of requirements is useful to distinguish spillover-related infection of urban residents exposed to infective sylvan vectors from genuine transmission cases by *Ae. aegypti* in the urban environment. Today, back-tracking of imported cases can be more readily performed, thanks to software improvement during the COVID-19 pandemic.

#### 5.5.3. Urban YF Outbreaks in the Americas after 1910

In the early part of the 20th century, urban areas were still considered an epidemic focus and thus “reservoirs”. After the discovery of sylvan transmission, the reservoir question shifted to sylvan environments. Also, the notion of a chronic carrier role of *Ae. aegypti* was abandoned, because after the end of an outbreak, no further cases developed in those urban areas despite perpetual infestation by this mosquito.

In South America, large urban outbreaks occurred in port cities from Cartagena, Colombia to Buenos Aires, Argentina. The last major urban YF outbreak in seaports in the Americas occurred in Rio de Janeiro in 1928 [191]. In this outbreak, out of 737 cases reported, 417 died [192]. Smaller outbreaks occurred in the interior regions of Brazil. Although the outbreak in the State of Acre has been traditionally identified as the last urban outbreak in Brazil, a question was raised for the record not meeting the aforementioned requirements for urban outbreaks [73,193]. Actually, the last better-documented urban epidemic in Brazil was recorded in Canbará, State of Paraná, Brazil in 1936 [194]. In this outbreak, the index cases were tree cutter and truck driver who hauled lumbers. Secondary transmission by *Ae. aegypti* in this outbreak was confirmed. The epidemic data in four countries (Brazil, Colombia, Bolivia and Peru) in 1934–1949 revealed that after 1934, 97.9% of the fatal YF cases occurred in the areas without *Ae. aegypti* [22]. The last very small urban outbreak in the Americas occurred in Trinidad in 1954 [105]. More recently, “urban outbreaks” have been reported in Santa Cruz, Bolivia (1998–1999) and San Pedro and San Lorenzo, Paraguay (2007–2008). However, all involved work-related exposure of urban residents to the virus or other activities in sylvan environments [193]. It is stressed that change in vector competence is not the reason for the disappearance of urban YF in the Americas.

#### 5.5.4. Urban Outbreaks in Africa

The traditional hypothesis that *Ae. aegypti* adapted to the urban environment evolved from its subspecies in sylvan environments in Africa [33,37] has been generally well accepted. Historically, it was David J. Lewis who conceived the evolution of the domesticated form of this mosquito from zoophilic, sylvan population of *Ae. aegypti* in East Africa, while he studied a mixed population of anthropophilic-zoophilic types observed in the Nuba Mountains of Sudan [195]. Co-habitation of zoophilic and anthropophilic types in East Africa was also discovered in Kenya later.

In Africa, the history of outbreaks of diseases retrospectively suspected to be YF is long. As for more reliable urban transmission in Africa, scarcity of reliable documents is a major problem. In Saint-Louis, Sénégal, outbreak became more frequent in the 19th century, as evidenced by the enactment of quarantine laws in 1822 and 1876 [196]. YF outbreaks occurred frequently in Gorée, a small island offshore of Dakar, as it served as the largest slave-trading center in Africa. The first urban outbreak in Nigeria was recorded in Lagos in 1864. A selected list of the events in the 20th century includes outbreaks in Lagos, Nigeria (1925–1926), Bathurst, Gambia (1934), Accra, Ghana (1926–1927), Ogbomosho, Nigeria (1946, 1987), Diourbel, Sénégal (1965), Luanda, Angola (1971, 1975, 1988 and 2015–2016), and Buchanan, Liberia (1995). The frequent outbreak in Luanda suggests that the YF corridor is opened more frequently there, despite the large size (currently approximately 2.5 million) of the population. It is of interest to note that it is a port city busy with human movement. Historically, YF-like outbreaks have occurred there as early as in the 17th century. In the 2015–2016 outbreak, the influx of a large number of overland migrants from the Democratic Republic of Congo was one of the factors that complicated the accuracy of epidemiologic statistics. As mentioned earlier, outbreak at seaports in Europe and North America ceased to exist after 1910. However, outbreak at seaport still occurred in the YF zone of Africa, such as an outbreak in Accra, Ghana in 1937 [74].

Length of outbreak in urban area: The length is usually much less than 1 year, ranging from 1 to 2 months in Sénégal to 11 months in Ethiopia [197]. In reality, these lengths rarely reflect “natural” length, because of quick intervention with vaccination and vector control. Outbreaks lasting more than one year were reported, but, based on the records alone, the possibility of the repeated introduction of the virus, each time from a different enzootic/epizootic focus to the same urban center, could not be totally eliminated.

Magnitude of outbreak: In the 1986–1987 outbreaks in three states of Nigeria, estimated numbers of cases and deaths were 120,000 and 24,000, respectively. The comparable figures in the 2015–2016 outbreak in Luanda, Angola were 7334 and 393, respectively [198]. However, because of communication and diagnostic problems, the data likely represented under-reporting. In other outbreak in the villages in Gambia (1978–1979), initial epidemiologic figures were low, but when they were more comprehensively studied, the estimated numbers of YF cases and deaths rose to 8400 and 1600, respectively [68].

## 6. Virus

With the exception of the introduction to the early history of YFV isolation (Section 6.1), the main objective of the remainder of this section is the search for the unique traits of YFV among four viruses transmitted by *Ae. aegypti*.

### 6.1. A History of Early YFV Isolation

Shortly after the determination of the mosquito-borne transmission of YF in 1900, William Welch of the Johns Hopkins University advised Reed and James Carroll a possible importance to YF investigation of the report of Friedrich Loeffler and Paul Frosch, who determined the filterability of the etiologic agent of foot-and-mouth disease in 1898 [199]. Carroll quickly confirmed filterability of the YF agent in Havana [200]. Aristides Agramonte of the Reed Commission attempted to isolate the etiologic agent but failed, despite the fact that he used rhesus monkey [94].

In 1927, the RF’s YF investigative group (principally Adrian Stokes and Alexander F. Mahaffy) in Yaba, Nigeria finally succeeded in isolating YFV from the blood of a Ghanaian patient (named Asibi) [201]. Shortly, Constant Mathis, Andrew W. Sellards, and Jean Laigret in Sénégal succeeded in isolating the second strain from a patient (named François Miyeli) [202]. In Brazil, Henrique de Beaurepaire Aragâo isolated the third strain from a Hungarian resident in Rio de Janeiro in 1928 [203]. While the history about the first isolation in Nigeria has been abundantly presented in many reviews and books, what has not been revealed until 1997 was the contribution of Oskar Klotz, a renowned Canadian pathologist.

Klotz had an experience with pathologies of YF-infected monkeys while he was in Sâo Paulo, Brazil to help establishment of the School of Public Health at the University there. Later, he served as a consultant for the RF specifically with respect to YF research in Yaba, Nigeria. When all initial attempts to isolate the etiologic agent of YF failed there, Klotz advised the RF staff to use Asian monkeys instead of guinea pigs and African monkeys. Frederick Russell, the Director of the International Health Division of the RF, however, rejected the proposal because he did not believe that *Ae. aegypti* would bite monkeys and because monkeys were too expensive. Eventually, he agreed to import Asian monkeys [14,99]. YFV was finally isolated [204], and the rest is history.

The second piece of the hidden history behind this is how the Asibi strain was identified as the cause of YF. At the time of isolation, little was known about the optimal condition for the long preservation of a virus. Some still believed in a myth that tropical pathogens would lose infectivity if kept in a cold environment. Preservation in 50% glycerol in a cold environment was employed by some, but the refrigerants used at the time (such as ammonia and methyl chloride) were dangerous. Under this difficult condition, Marchoux and Simond had to ship cages of mosquitoes (which they thought infected) from Rio de Janeiro to Paris. Accordingly, at first, only fixed tissue specimens were shipped from Yaba to Toronto, where Klotz histopathologically confirmed YF etiology, based on the observation of Councilman’s bodies unique to YF [205]. The importance of the Councilman’s discovery was recognized nearly 8 decades later as the first report of apoptosis.

In 1928, Andrew W. Sellards successfully shipped blood specimens and tissue blocks of infected liver in specimen tubes wrapped in wet ice and salt from Sénégal to London, a journey that took 12 days by ship. Shortly, it was found conclusively that either freeze-drying or cold temperature in the refrigerator of ship actually would preserve infectivity much longer [206]. Also, an effective and safe refrigerant (“Freon”^TM^) began to replace old refrigerants. Thus, in his second visit to West Africa in 1928, Klotz could bring home the Asibi strain. He conducted a neutralization test using antiserums from human cases and confirmed the etiologic agent [101]. Only then could RF officially announce isolation of the YF agent.

### 6.2. Differences between African and American Isolates of YFV and Virus Characterization

Delfin H. Clarke of the RIMR differentiated African strains from the New World strains based on antigenicity by conducting a cross-absorption of immune serums prepared with homologous and heterologous strains [207]. The full genome sequencing of YFV by Charles M. Rice (later, Nobel Laureate) [208] opened a new chapter in arbovirus studies. The first electron microscopic study of YFV was performed by Reginald L. Reagan [209].

Evidence of genomic differences between the African and American strains of YFV was reported by Vincent Deubel and his colleagues [210]. Genotyping study revealed either four or two types in Africa and two types in the Americas [211,212]. A phylogenetic study conclusively determined that the American strains were indeed derived from West Africa. While this direction of the viral dispersal has been well established, historical documents reveal that the home coming of YFV from South America to West Africa was also possible. For example, in an outbreak in Saint-Louis, Sénégal in 1878, it was believed that the epidemic was caused by the arrival of ships from Brazil [196]. Since slave ships repeatedly came back to Gorée, located south of Saint-Louis, to pick up slaves, this possibility may not be easily dismissed.

### 6.3. The Rate of Mutation and Phylogenetic Tree Topology

For the identification of the uniqueness of YFV among four arboviruses transmitted by *Ae. aegypti*, the rate of mutation is used for comparison. Then, phylogenetic branch topologies are examined to find out if isolates from the urban environment are clearly segregated from the isolates from other environments.

To minimize variation among reports due to differences in viral lineage, dispersal history, the quality of samples selected, human and other factors, the data primarily based on the studies in the South America are arbitrarily selected for comparison among three flaviviruses transmitted by *Ae. aegypti*. The rates determined on the E gene among DENV2, YFV and ZIKV were 5.8–7.0 × 10^−4^, 2.1–3.3 × 10^−4^ and 8.12–9.71 × 10^−4^ substitutions/site/year, respectively [213,214]. It is probable that the very low rate of YFV suggests that YFV is not involved in many replication cycles involving host per given period, compared with the other viruses. Directly comparing these rates with ZIKV rate, however, is complicated by the evidence of recombination with a breakpoint in the E gene. For this virus, which geographic strains to choose for comparison (African, Asian, or American) also contributes to the variation in the comparative significance.

In the phylogenetic tree of DENV2, urban strains are clustered in a basal branch segregated from the ancestral branch [215]. On the other hand, no such a basal branch segregation is found in the phylogeny of YFV, confirming that all epidemics or epizootic strains are direct descendants of the sylvan sources by in situ evolution [152]. This corresponds well with the fact that outbreaks of YFV in transitional and urban environments are usually short. On the other hand, DENV2 more often persists far longer if the size of the population of urban area infested by the vector exceeds one million (or minimal 150,000) [216]. As for ZIKV, a major problem is the difficulty of segregating strains into sylvan or urban. Accordingly, no such a branch separation, as observed in DENV2 phylogeny, was found [217,218]. This lack of branch separation also applies to CHIKV phylogeny.

### 6.4. Host Adaptation (or Increased Host Resistance) Concept Regarding the Long-Term Relationship between the Pathogen and the Host

In the traditional concept (so-called “conventional wisdom”), the long-term relationship between the infectious agent and the host would eventually evolve to mutual coexistence. The three mechanisms considered were (i) a reduction in virulence of the pathogen; (ii) increased resistance of host measured at population level; and (iii) the interaction of (i) and (ii). In this review, only (ii) is discussed. Also, this review is focused only on the very early phase of this long host–parasite relationship when pathology in host is still severe. It is also emphasized that this is a population-based concept and not applied to individual cases.

The following two observations are useful for arboviruses. In the first relationship, in a given area (indigenous area) where a pathogen (indigenous pathogen) and a host (indigenous host) coexist, this is an indigenous–indigenous relationship. In the early phase of this relationship, the pathology and mortality in the indigenous host are considerable.

In the second relationship, populations of host identical to the indigenous host or of a different species taxonomically very close to the indigenous host disperse (or are imported) to the indigenous area. These hosts have never been in contact with the indigenous pathogen before because of geographic separation. When this happens, pathology in both kinds of hosts is severe at first. In reverse, when an indigenous pathogen disperses to new territories where the populations of those naïve hosts occur, the same phenomenon is observed. This is an indigenous–exogenous relationship. Many examples of the second observation are known among arbovirus [219].

The unstated ideal requirements for this concept are that the long history of the pathogen–host relationship has been reasonably well established and that the unrelated influx of pathogen or host to the area of interest has not occurred. As common to many sweeping generalizations of biological phenomena conceived in the early period of scientific development, multiple caveats were discovered by later generations of scientists regarding this conventional wisdom [220]. Host changes in the composition, the size of the host population, the change in contact rate between host and pathogen, and the pathogen’s evolutionary (or genetic) change in host are just some of the reasons. Thus, the currently blanket application of the “conventional wisdom” for many infectious diseases is not feasible.

However, the mechanism (ii) above is still applicable to YFV, as shown by the lower fatality rate (CFR) among the residents of Africa vs. higher CFR among residents in South America, even though a possibility of genetic difference in virulence factor between YFV populations in two continents as another explanation cannot be entirely ignored at this moment. When sizable numbers of Europeans (including groups of visitors, immigrants, and soldiers) moved to the YF zone or other locations in Africa or in the Americas repeatedly affected by YF, outbreak often erupted soon among the newly-arrived, while recognizable cases among the natives occurred only sporadically. In the U.S. too, mortality was higher among newly-arrived immigrants than in long-time residents (both white and black) [221]. This clear difference in mortality was pronounced in New Orleans, where new immigrants kept arriving. Decimation of Napoleon’s army in Saint Dominique (now Haiti), a legendary story of the “white man’s grave” in Africa, and high mortality among British Naval servicemen in the Caribbean garrisons are other examples [222,223,224]. During the 1928 outbreak in Rio de Janeiro, foreigners were conspicuously more infected (78%) than Brazilians (22%); and CFRs (%) for foreigners and Brazilians were 62 and 33, respectively, despite the fact that foreigners made up only 20.8% of the population of the city at the time [191]. Lower CFRs of the NHPs in Africa, in contrast to higher CFRs in the NHPs in the Americas, is an additional supporting observation.

One important peculiarity of YF is that even in the indigenous Africa, where there is an indigenous–indigenous relationship with humans, the prevalence and CFR are still considerably high. This likely reflects YFV’s trait, very high CFR. A possible other explanation is that an equilibrium in the indigenous–indigenous relationship least pathogenic to the host has not been attained yet for YFV.

Unlike YFV, with a past history of dispersal that is most reliable because of the abundance of historical records and the pathognomonic syndrome, for the other three viruses, evaluation of the applicability of the conventional wisdom is difficult. Although their birthplaces were established in molecular phylogenetic studies, reliable data on subsequent viral dispersal are scarce. Furthermore, even if historical records indicate the probable occurrence of a disease in one region of the world but subsequently virus disappeared before the age of virus discovery, these places are probably not recognized either as the origin or the place of dispersal, since no virus specimen is available from the early periods. As an example, CHIKV most likely dispersed from Africa to the Americas in 1826–1828, but the lineage presumably became extinct. This virus also dispersed multiple times from Africa to Asia at least since the late 18th century and established enzootic transmission there [225,226]. But even the exact current boundary of its enzootic zone in Asia is unknown.

As for DENV, although origin in Asia has been a dominant theory [227], a more complicated theory of dispersal was recently proposed [228]. The discovery of cosmopolitan lineage of DENV2 further complicated the history of dispersal [229]. In addition to sharing of disease syndrome with other viruses, the lack of reliable historical records rendered it difficult to determine exactly how and when DENV dispersed to the Americas [230]. Some even speculated a possibility of dengue dispersal from Africa to the Americas by the slave trade. ZIKV shares some of the problems of both CHIKV and DENV. Thus, without a reliable historical record of dispersal, the concept of host adaptation cannot be evaluated for those viruses.

### 6.5. Virulence and Trade-Off Theory

This trade-off theory was developed as an alternative to the aforementioned adaptation concept. It has little to do with other trade-off theories for arboviruses regarding gain or loss of fitness to either vector or host. In this review, CFR (expressed in %) is used as an indicator of virulence. According to this hypothesis, the higher the virulence, the less transmissible the infectious agent.

However, the selected denominator for YF CFR is highly variable among countries. Also, whether or not patients of mild symptoms actually see a physician varies considerably among countries due to differences in health care system, efficacy of reporting system, economics, cultural tradition and other factors. In the 1978 YF outbreak in Nigeria, two estimated CFRs were 50 and 75. In Sénégal, CFRs ranging from 16 to 28 were reported. However, when mild cases were included, it dropped to 12. When CFRs from all outbreaks in Africa (1975–2020) were compared, it ranged from 86 in Nigeria to 10 in Ghana, while mortality ranged from 0.1/100,000 in Nigeria to 2200/100,000 in Ghana [197]. In 2013, the rate for Peru and Colombia combined was 65.2, whereas it was 33.6 during the 2016–2019 outbreak in Brazil [231]. Despite the problems of variations in the criteria applied in the selection of denominator among countries, a meta-analysis revealed a higher CFR (47) in South America and a lower CFR (36) in Africa [232].

When compared with the CFR of YF, the corresponding values (in %) in the Americas for CHIK, dengue, and Zika fever were 0.032, 0.046, and 0.005, respectively [132]. Conversely, the prevalence (total number of patients) per outbreak is far greater in these three viruses than in YF, even after discounting the accuracy of epidemiologic statistics because of under-reporting, which is applicable to all four diseases.

Many examples of infectious diseases that do not fit well with this trade-off theory were reported [233]. The definition of “virulence” has also been a source of debate. Nonetheless, this hypothesis may be still applicable to YF, as explained in Section 6.7.

Although the exact cause of the huge difference in virulence (measured by CFR) between YFV and other three viruses transmitted by *Ae. agypti* is still only partially understood, traditionally the uniqueness of the YFV affinity to hepatic tissue has been speculated as the principal cause [234]. As virulence typically entails a combination of the impacts associated with viral infection of host, the involvement of multiple viral genes is suspected. Thus far, the involvement of C, E, NS1, and 3′ UTR of YFV has been found important. 3′ UTR alone is involved in genome cyclization, replication, transmission, and virulence. Thus, this area of research needs to be further explored to explain more clearly the very high CFR and the limited global dispersal of YFV among the four viruses transmitted by *Ae. aegypti*.

### 6.6. The Importance of Epizootic Outbreaks in Viral Transmission in Transitional and Urban Environments

Although YFV transmission in three different environments is typically illustrated as shown in Figure 1, if the origin of the viruses that initiate an outbreak in either urban or transitional environment (called “seed viruses” for convenience) is emphasized, another distinction among the four viruses transmitted by *Ae. aegypti* emerges. Figure 2 shows flows of YFV and DENV from one environment to another, beginning with the original enzootic cycle (black arrow) for YFV or an urban cycle (black arrow) for DENV. Only the YF and dengue outbreaks that have an origin and dispersal were well recorded in modern times were used for the drawing of Figure 2. The chain of flow shown in Figure 2A for YFV typically lasts only for a short period. Thus, the number of transitional or urban environments involved is small. It is noted that seed viruses from an epizootic cycle initiate transmission in the first transitional environment. If further spread occurs, transmission in either the second transitional environment or the first urban environment is initiated by the seed viruses from the first transitional environment. Thus, regardless of the difference in environment, outbreaks in those environments essentially depend on the original seed viruses from the epizootic focus.

On the other hand, DENV transmission is almost totally independent of such a reliance on seed viruses from sylvan environment, as shown in Figure 2B. The dispersal of DENV from sylvan to transitional environments apparently occurs infrequently [235,236]. However, the dispersed virus from the sylvan cycle is probably not required to function as a source of seed viruses at least in the urban environment, because urban cycle is self-sufficient to perpetuate BT. Figure 2B also illustrates that DENV dispersal occurs between urban and transitional environments. While transmission in large urban areas may last for several years or longer, at this moment, little is known about how long transmission in transitional environments persists. In contrast to YFV in transitional environments (where vectors other than *Ae. aegypti* are typically involved), the vectors in dengue transmission in urban and transitional environment are *Ae. aegypti* and most likely *Ae. albopictus*. Accordingly, the common practice of characterizing dengue as exclusively an urban disease was found inappropriate [237].

As for CHIKV, it is known to have two zones (or enzootic foci) of sylvan transmission in Africa demonstrating synchrony. However, other than that, establishing the concept of seed virus is generally difficult in Africa and Asia-Pacific. Typically, the virus suddenly causes an outbreak, quickly disappears and reappears later at unexpected locations at an unpredictable time [238,239]. The longest length of persistence (confirmed by virus isolation) in a fixed urban area was 3 years recorded in the metropolitan Bangkok in 1967 [240]. For ZIKV, except for the reliability of its birth place in Africa and subsequent spread to Asia and more recently to the Americas [241], little is known about the boundaries of enzootic foci. Furthermore, for both CHIKV and ZIKV in the Americas, it is too early to determine if they will eventually establish enzootic transmission there.

Accordingly, the four viruses may be colloquially classified as follows: YFV/sylvan DENV and urban DENV are sedentary home-based viruses with residence in sylvan and urban environments, respectively. The difference between YFV and DENV is that the former tends to change sylvan home address frequently, while sylvan address of DENV remains unchanged; and urban DNV stays in one address for many years. On the other hand, CHIKV and ZIKV are largely “nomadic traveling viruses”, because we know neither their provenance nor destination. Also, we do not know where they are during the inactive period, except in a few presumed enzootic foci in Africa. Simply put, mostly they do not have an established residence.

### 6.7. Synthesizing the Uniqueness of the Requirements for YFV Transmission with an Emphasis on Geographic Distribution of YF

The four viruses transmitted by *Ae. aegypti* have attracted attention of many scientists for the epidemiologic traits they share. Finding the shared and unshared molecular traits of these viruses is clearly one of the strategies to unravel the uniqueness of YFV [242].

In this review, multiple epidemiologic traits that have been known for many years are re-evaluated to find out the differences among four viruses. Based on the data in the Section 6.2, Section 6.3, Section 6.4, Section 6.5 and Section 6.6, the following combination stands out for the uniqueness of YFV: pathognomonic syndrome characterized by jaundice and hemorrhage [243], high CFR, the lack of dispersal of enzootic focus out of the traditional YF zone, and total dependence of transitional/urban environments on the “seed viruses” from epizootic focus in sylvan environments.

In this respect, it is important to re-examine two exceptionally prolonged YF outbreaks, the 1934–1940 outbreak in South America and the 1948–1957 dispersal in Central America [244,245]. These two outbreaks can be interpreted as nature’s failed attempts to expand the YF zone or establish enzootic foci in a new territory. Since the first one was analyzed earlier [22], only the second one is commented on. Jorge Boshell Manrique of Colombia, who was well experienced in the investigation of sylvan transmission in South America, identified the following facts regulating the epizootic outbreaks in Central America: diversity and population size of NHP fauna, rainfall or length of the dry period, and the efficacy of vectors in maintaining virus [246]. He noticed a limited diversity of NHP fauna in Central America compared with its richness in South America. He speculated that as populations of NHPs (*Alouatta* and *Ateles*) were decimated, prolonged outbreak was sustained only by the populations of more resistant *Cebus* monkeys. Also, he noted that Mexico-Guatemala border area, where the dispersal of the outbreak stopped, corresponded to the northern limit of the distribution of many NHPs. In fact, both howler and spider monkeys in southern Mexico have been on the endangered animal list for protection. Boshell still thought that the rain forests along the Atlantic were more favorable for transmission than the deciduous forests along the Pacific. Interestingly, in South America, enzootic foci are all located east (but not west) of the Andes, even though historically small outbreaks of unknown provenance occurred before 1930 in the communities along the Pacific coast of Peru (Callao, Piura and Lima) and Ecuador (Guayaquil), strongly suggesting the importance of rainfall.

If this interpretation of the 9-year outbreak in Central America is useful for discussion, it points out that apparently even the favorable ecological conditions in the Atlantic side of Central America did not meet all unique requirements for the establishment of enzootic focus or the YF zone, despite the occurrence of susceptible NHPs and sylvan vectors there. Then, the regions surrounding the Caribbean provide an opportunity for a comparative ecological study to unravel the mechanisms of YFV persistence in sylvan environments. In the islands of St. Kitts/Nevis and Barbados, African green monkey populations are approximately 60,000 and 14,000, respectively, but primary sylvan YF vectors do not occur. Naturally, enzootic focus has never been recorded. In Central America, YFV failed, despite NHPs and sylvan vectors. In Trinidad, where NHPs and sylvan vectors occur, at least short-term persistence has occurred repeatedly over a long period in the past, most likely because of its close proximity to South America which probably served as the repeated source of seed viruses. In South America, it has been firmly established.

Then, the following three recent reports deserve mention, regarding the unique requirements for YF. In the first report, correlations between entomologic and environmental parameters for a transmission in a state of Brazil (where YF activity was absent for several decades) were evaluated to determine the combinations of parameters that either favor transmission or do not favor it [170]. In the second report, impacts of the variables of biologic, topographic, and climatologic factors (even including wind) were emphasized [247]. Both the first and second reports rely on algorithm-based modeling and are more useful for identifying transitional environments. In the third report, the objective was to identify spatial patterns and the relationship between key geographic and environmental factors for enzootic/epizootic transmission in South America and Trinidad, with a focus on human infection [248]. If more research is conducted for both kinds of approaches, they will provide invaluable sources of environmental requirement information for YFV persistence.

A similar gradient of the occurrence of YF outbreaks is also found in East Africa: countries surrounding the YF zone but without YF where *Ae. africanus* may or may not occur but where multiple other vectors and many species of NHPs occur (Malawi, Mozambique, Rwanda, Tanzania, Zambia and Zimbabwe); countries of infrequent YF outbreaks but where multiple species of vectors including *Ae. africanus* and NHPs occur (Kenya and Ethiopia); countries where outbreak is more frequent and multiple vectors including *Ae. africanus* and NHPs occur (Sudan and Uganda). The analysis in East Africa, however, is complicated by the fact that old distribution records and taxonomic position of some species (such as *Ae. taylori*) in several countries have been found questionable; and new species was proposed. Regarding the infrequent outbreak overall and the absence of urban transmission by *Ae. aegypti* in East Africa, the importance of the unique ecosystem and the possible involvement of the indigenous viral genotype were proposed [249], while others suspected lower competence of the vectors there.

Thus, the above discussion makes the application of the trade-off theory to the failure of the establishment of YF foci in Central America feasible. In combination with the total dependence on sylvan epizootic outbreaks for seed viruses in transitional and urban environments, this failed history may partially explain why the YF zone has not expanded beyond Africa and South America in nearly 150 years after the end of slavery. When the earlier map illustrating distribution of seropositivity to YFV was prepared in early part of the 1940s, serologic survey was still incomplete in parts of East Africa and southern parts of South America (Figure 3A). Once the survey was completed since, these parts were added. Thus, basically no significant expansion of the YF zone has occurred over nearly 8 decades (Figure 3B). In contrast, CHIKV, DENV, and ZIKV expanded their territories in three continents and the islands in the Pacific. Unquestionably, as in the assessment of any new hypothesis, further research is still necessary from all perspectives before reaching the final conclusion for the following reason. The impact of the requirement of YF vaccination certificate for certain international travelers, for example, could have been significant for the difference in dispersal, since no similar requirement has ever been implemented for other three viruses because of lack of vaccine and other practical difficulties in controlling their international spread. A recently approved CHIKV vaccine may change the pattern of this viral transmission in the future, however.

## 7. Relevance of Geologic Events in the Understanding of the Evolution of Flaviviruses, Vectors, and of Vector-Borne Viruses as Well as the Geographic Dispersal of Arboviruses Including YFV

When the mechanisms of the natural maintenance of YFV and of the emergence of YF outbreaks are poorly understood, as today, all possibilities should be put on the table for discussion. The application of the Continental Drift theory proposed by Alfred Wegener is an example.

### 7.1. The Evolution of Flaviviruses

Radical change in the timeline of the evolution of flaviviruses became possible after the seminal discovery by Mang Shi and his colleagues of a large number of new flavi-like viruses bearing unique flaviviral signals [250]. These signals are collectively called “flavivirids”, such as endogenous viral elements, motifs, genes and others [251].

The findings in the latest phylogenetic study of flavivirids [251] are summarized as follows: (i) multiple lineages of viruses initially evolved in major groups of primitive organisms in a complex series of events; (ii) the dominant primordial organisms in which flavivirids initially evolved were probably aquatic cnidarians (i.e., jelly fishes). From these organisms, flavivirids further evolved in aquatic arachnids (i.e., mites), terrestrial ticks and mosquitoes in that order. The following additional data by others further support this latest finding. The evolution of mosquito-borne flaviviruses from tick-borne viruses was deduced earlier and replication of West Nile virus in tick cell cultures was speculated to be a vestigial sign of the past association with ticks [252,253]. Interestingly, the origin of the Family *Togaviridae* was also determined to be aquatic [254], and a unique strain of mosquito-borne Semliki Forest virus (an alphavirus) was found to replicate in tick cell cultures [255].

According to this recent finding [251], in the early phase, the dominant aquatic environments were both fresh water and saline water; (iii) the suspected modes of the sequential transfer of flavivirids from one group of invertebrates to another or to vertebrate hosts were via intracellular exosomes, oral ingestion (including cannibalism), scavenger feeding in close contact with host ultimately leading to the development of mechanical transmission, and finally blood feeding in that order; and (iv) blood feeding by different groups of arthropods evolved more than 20 times as a convergence. However, blood feeding was gained and secondarily lost, depending on viral lineage. Conversely, probable transition of originally vector-associated flaviviruses to vertebrate viruses is strongly suspected in three members in the vector-borne flavivirus branch. This is based on the fact that these vertebrate viruses without a vector (Entebbe bat, Yokose, and Sokuluk viruses) still replicate in mosquito cells in vitro [253,256]. These changes are reflected in the complicated, paraphyletic branching topology of flaviviruses leading to the evolution of strictly vector-borne group. Accordingly, the authors of latest report [251] concluded that the evolution of a variety of flaviviruses with a diversity of biologic characteristics could be best understood on the basis of the Continental Drift. They also concluded that the current understanding of the timeline of the evolution of flaviviruses (only within 10–100,000 years ago [257,258]) needs to be revised drastically. Naturally, the topology of the phylogenetic tree of genus *Flavivirus* based on full-length genomes [259] also needs to be revised.

### 7.2. The Evolution of Mosquitoes

In the second recent publication, the evolutionary history of mosquitoes was re-examined [260]. The study revealed that the subfamily *Anophelinae* evolved in the early Triassic Period (217 MYA; range 250–188 MYA) from the common ancestors. The subfamily *Culicinae* then evolved, generating multiple genera such as *Uranotaenia*, *Toxorhynchites*, *Sabethes*, *Haemagogus*, *Aedes*, and *Culex*. Two genera (*Aedes* and *Culex*) are not monophyletic, suggesting a complex evolutionary history behind. Paralleling the evolutionary process of flavivirids above, the mode of the feeding behavior of mosquitoes changed from free living to blood-feeding (in the adult stage) upon the development of circulatory system in vertebrates. It was concluded that this evolutionary process in mosquitoes was also best explained by Continental Drift, but it was stressed that mosquito phylogeny was further modified after the Continental Drift through variation in the relationship with host groups.

### 7.3. Additional Information

More available data support the above two conclusions. First, flaviviruses were found in anopheline mosquitoes, which are now considered the earliest mosquito group that emerged [261]. In the cell culture of nonblood-feeding mosquitoes of Genus *Toxorhynchites*, which are not closely related to Genera *Aedes* and *Culex* but which are still members of the subfamily *Culicinae*, YFV and other vector-borne flaviviruses nonetheless replicate well, revealing a hidden but shared trait in the subfamily [262,263]. In fact, infection in the larval stage has been confirmed in several other arbovirus families; and data on the replication of vector-borne viruses in aquatic organisms (i.e., fishes) or in their cell cultures are extensive [264]. In many aquatic environments in Africa (including “dambo” in East Africa), wildlife (birds, amphibians, mammals, and others) share these environments with aquatic invertebrates, including mosquito larvae belonging to multiple genera. As cannibalism and natural death occur, releasing viruses, opportunities for mosquito larvae to ingest a variety of viruses are plenty. Thus, these aquatic environments provide an ideal condition for virus mixing. This is comparable to the environment in southern Asia where new antigenic varieties of influenza virus are constantly generated via genetic reassortment in the environments where birds, pigs and other animals congregate, a favorable condition for virus mixing. This may explain why phylogenetic incongruence involving insect-specific and vector-borne flaviviruses and between *Aedes*-born and *Culex*-born flaviviruses is common [265].

### 7.4. Understanding the Uniqueness of the Current Geographic Distribution of Viruses Including YFV

As for unraveling the mechanisms involved behind the puzzling patterns of geographic distribution of vector-borne viruses, it was Charles H. Calisher who pioneered linking early events in geologic periods many millions of years ago to secondary modifications that happened after the completion of Continental Drift [266]. Thus, the viruses adapted to migratory birds, bats and highly mobile terrestrial animals tended to show more secondary global or transcontinental distribution after Continental Drift, while those adapted to vectors or hosts with limited mobility were restricted to “nidi” in limited regions. Because blood feeding was gained and lost more than once among flavivirus groups, understanding the history of the feeding behavior of any vector group involved was also critically important for properly understanding the pattern of geographic distribution. Subsequent research by others further identified the secondary modification of viral distribution after the rise of human communities. Such human factors include domestication of animals (animal husbandry), deployment of animals for transportation in long-distance trade, and many others (including the regrettable history of the slave trade). The viruses that adapted to migratory birds, mammals (including bats and aquatic migratory mammals), and others, as well as viruses of their ectoparasites (in particular ticks, mites and lice), demonstrate a tendency of more global, transcontinental or transoceanic distribution.

The above discussion is useful for more clearly understanding the pattern of geographic spread of arboviruses, a composite of viruses belonging to multiple virus families which were artificially bundled together only because of a convergent trait, BT. Should we focus on the long evolutionary history of each virus lineage even going back to the geologic period (choice 1)? Or should we focus on blood-feeding on humans by domesticated *Ae. aegypti* in the urban environment (choice 2)? It is noted that urban breeding and dependence on humans as source of blood meal by mosquitoes (including *Culex pipiens*) developed only recently when human communities began to evolve (approximately 10,000 B.C.E.) [33]. Responding to such a question is difficult because viral dispersal entails multiple factors, the length of association with vector or host being only one of them. Nevertheless, when we focus attention to YFV, in the choice 1, two players of BT (NHPs, and sylvan vectors) are found to be both absolutely indispensable for the survival of this virus. On the other hand, in the choice 2, both human and *Ae. aegypti* are dispensable. Accordingly, as far as YFV is concerned, the data thus far available favor consideration of the choice 1 rather than of the choice 2. On the other hand, this logic is not applicable to the other three viruses because of the paucity of reliable information about the past history of dispersal (Section 6.7). Conversely, choice 2 fits DENV. Thus, the prevalent practice of studying YFV transmission based on DENV transmission, ranging from mathematical modeling to determination of reproduction number (R_0_), is questionable.

## 8. Issues Surrounding Research

Many problems experienced by researchers in the field and laboratory provide a rich source of important lessons generally neither taught in virology classes at academic institutions nor described in most review articles. However, probably the far more important source of problems is the arbovirus concept itself.

### 8.1. Basic Sources of Problems

Studying the dynamics of arboviral transmission under natural conditions to secure evidence to fill in blanks in the BT cycle is an arduous and very difficult task. Four kinds of problematic issues have been found to be the sources of this difficulty.

First source of problems: Although schematic cyclic presentation of BT (Figure 4A) superficially looks simple and easy to understand, what such a figure does not reveal is enormous complexity of the numerous interactions involved in each step. Each arrow in the cycle (Figure 4A) represents a directed process, beginning with a particular status of virus, originally with virus in infective vectors and ending with infected hosts. This must be conducive to completion of the cycle. When each arrow is more closely examined, the involvement of numerous and complicated interactions among parameters becomes evident (Figure 4B). In Figure 4B, the parameters peculiar to the virus are not shown, to make the figure simpler. Some of the parameters enter the process only after prior modification by physical factors of the environment (such as ambient temperature and rainfall) (Figure 4B). As evident, collecting all necessary data in the field just to complete one arrow of a BT cycle is challenging. When multiple species of vectors and hosts are involved in transmission, as in the sylvan transmission of YFV, complication is further magnified. The changes in the status of vectors or hosts and the sequence of events that follows, as well as the synchrony among the virus, the vector and the host to meet, are then integrated to complete one BT cycle. Synchrony itself is a collection of multiple parameters and factors.

Fully acknowledging the depth of the difficulty in the field, the advisory group of the WHO listed a number of guidelines to help the workers and encouraged laboratory simulation to partially offset the difficulties in the field [1]. It is noted that, for a realistic reason, the arbovirus concept by the WHO did not stipulate that vector-host alternating BT cycle must be faithfully repeated consecutively, allowing freedom to apply a combination of other transmission modes [267]. In some institutions, a set of internal working definitions, requirements, criteria, and other specifications were generated for a smooth communication within each group. The definition of vector competence, the list of qualifications to be primary host or vector of arbovirus, and the requirements for a reservoir are such examples. Naturally, in the absence of a universal protocol or definition, variations in laboratory experiments or in the criteria for judgment evolved. When closely examined, even some reasonably well-established examples of BT are still at least partially based on a variable degree of extrapolation [148].

Second source of problems: Scientifically questionable principles or requirements, the lack of a clear definition, the ambiguity that allows multiple interpretations and the miscellaneous weaknesses of the arbovirus concept have been serious impediments to the progress in this branch of virology. For example, contradiction (such as nonexistence of vertebrate reservoirs of RNA arboviruses) negatively impacts the validity of research conducted on the belief that arbovirus diseases are zoonoses. Unfortunately, arbovirus research communities worldwide lack a global organization and a platform comparable to the International Committee on Taxonomy of Viruses (ICTV) to discuss any problems, correct discrepancies and revise the definitions, principles, protocols and others as necessary. This is an urgent and critically important subject that requires an immediate action.

Third source of problems: Reduced manpower and support is a serious issue. As early as in the 1950s, signs of the problem were recognized [268], but the shift to laboratory research accelerated in intensity in the 1980s. It was mostly a consequence of the rise in popularity of new branches of science (in particular molecular virology and applications of computer science in biologic research) and associated shift in manpower, research priority and funding. As a result, well before the basic field data necessary to complete BT were secured, shift in emphasis or interest to new branches in virology was already in motion among younger generations of researchers. As an example, decline in field research was noted in the contrasting number of hosts of YFV examined for the seroprevalence and other types of studies, approximately 13,000 in 1930–1950 vs. only 1000 in 1980–2008 [249]. The negative impact of the depleted human resource for field research has been considerable [269,270]. This is troubling because molecular virologic research and traditional field research are mutually beneficial and complement each other [271].

Fourth source of problems: The absence of the emphasis on voucher system in the arbovirus concept. One of the most neglected subjects of infectious disease research in modern times is voucher specimen system. Though being quite different from the practice of type specimen deposition required by the International Commission of Zoological Nomenclature, deposition as voucher specimens of the vectors and hosts (previously described taxonomically by someone else) newly discovered for the first time to be involved in the transmission of infectious agents is equally important. When a virus switches vector or host as in spillback or when it disperses to a new territory and infects different species of fauna, it is an important opportunity for voucher deposition. This system ensures reproducibility in disease transmission investigation, by providing an opportunity for other investigators to examine the specimens.

As a result of near total neglect in modern times, it was reported that most recent arthropod research in this category falls in the domain of “nonreproducible science” [272]. In Africa, taxonomic problems of YFV vectors have risen in the past and some still remain unresolved. In South America, a few *Aedes* mosquitoes hitherto unrecognized have been proposed to be possible vectors of YFV. However, it is not clear from the literature [273] if those mosquito specimens were actually deposited as voucher specimens. Voucher specimens of hosts are also important in virologic research. In the recent COVID-19 pandemic by SARS-CoV-2, the failure to secure host voucher specimens (the presumed earliest sources) was pointed out as one of the major sources of problems for identifying the viral origin [274]. On the other hand, the HIV-1 genome extracted from formalin-fixed tissue specimens collected in the Democratic Republic of Congo in 1966 revealed a crucially important early history of the evolution of this virus well before the pandemic that began in the early 1980s [275]. Similar archival studies for the early YFV genotype may be possible if a sufficient number of fixed tissues have been preserved. It should be remembered that in Brazil, at least in the early part of YF research, a system of preserving voucher host specimens was initiated by the Ministry of Education of Brazil in collaboration with the RF, and the specimens were deposited in the Museu Nacional, Universidade Nacional do Rio de Janeiro and Smithsonian National Museum of Natural History in the U.S. [274]. Thus, it is important to revive the old tradition for YF research.

### 8.2. Puzzles/Unknown

Two puzzles and one unknown are elaborated below.

(i)Sylvan transmission in Muzo, Colombia: Muzo is a famed emerald mine town in the Department of Boyacá. YF outbreaks occurred there multiple times. Investigators discovered no evidence of the existence of either *Ae. aegypti* or NHPs. Only humans were the hosts. An obvious question raised was how YFV could persist if humans were not there. The repeated introduction of the virus by vectors to Muzo from other enzootic/epizootic foci and repeated introduction to Muzo by infected humans were two possible answers [167]. How YFV is maintained in the enzootic/epizootic foci year after year in the absence of NHPs was another puzzle. This is an enormously important question. Yet, research activity in the forest has dropped sharply in recent decades, except in Brazil and Sénégal.(ii)Why YF is absent in Asia: The warning of Manson over possible catastrophe in Asia if YF is introduced to Asia as a result of the completion of the Panama Canal [127] was instrumental in initiating a chain reaction to adopt preventive measures. As a result, quarantine inspection, ship fumigation, vector control at seaports, and a wireless international communication system were implemented in Asia. Through diplomatic negotiation, Anglo-Egyptian Sudan (now Sudan and South Sudan) was designated as a “buffer zone” [73].

Several theories have been presented for this YF enigma in Asia. However, no consensus has ever been obtained thus far. This writer holds an opinion that, despite sharing the mode of urban transmission by the same vector (*Ae. aegypti*), YFV has a set of unique characteristics. One of them is YF’s total dependence on epizootic transmission for an urban outbreak to occur. Accordingly, it is proposed that unless YFV establishes a stable enzootic zone or focus somewhere in tropical Asia, the possibility of autochthonous urban transmission derived from epizootic focus is nil. Then, the only expected YF problems in Asia are imported cases and limited *Ae. aegypti*-borne autochthonous urban outbreaks derived from imported cases. Thus, identifying competent primatophilic sylvan vectors, determining the actual roles of Asian monkeys in YFV transmission, and investigating other biotic and abiotic environmental conditions required for sylvan transmission in Asia are key to providing a definitive answer to this puzzle.

The other answer to this puzzle somewhat relates to “anophelism without malaria”, a phenomenon known in Europe [276]. This phenomenon relates to the disappearance of malaria from parts of Europe (where the disease had once existed at least for multiple centuries), despite the occurrence of malaria vectors. The decline in the infected *Anopheles* vector below the minimal population necessary to maintain malaria parasite persistence, a concept conceived by Ross [63], appears important. It is also possible that the disappearance of malaria was a result of a combination of a variety of environmental changes leading to loss of mosquito habitat, vector control and other forms of improved sanitation in rural communities. The absence of YF in Asia, on the other hand, refers to absolute nonexistence in history but not to disappearance. It has been dubbed “aegyptism without arbovirus” [277].

(iii)Possible return of YF to the countries in temperate climate: A similar question raised recently is a possibility of the return of YF to Europe (mostly around the Mediterranean) and to southern regions of North America where climate is warmer [278]. If this is still possible, a serious question is immediately raised about the validity of the proposed scheme of YFV transmission in the Section 6.6. The question concerns how extensive urban outbreaks in the locations far away from the sylvan environments in the YF zone in Africa or South America were possible in old days. Before arriving at an answer, it is first necessary to determine exactly how such a large epidemic occurred frequently before 1910 in the first place. Unfortunately, Hosack was the only one who recognized the importance of the abundance of mosquitoes during YF outbreaks [15]. He even did not know the name of the mosquitoes. Naturally, little is known about exactly how many infected passengers, infective vectors and uninfected mosquitoes disembarked at the ports of entry. All these blanks make estimation of the quantity of seed viruses introduced to port communities impossible.

Although the lack of these data is discouraging, nonetheless, the following information gathered from reading old literature provides something useful, even though it is also largely speculative. One such possible explanation is that the outbreaks followed arrival of multiple infected persons from the tropics, which were always accompanied with a simultaneous arrival of a sizable number of *Ae. aegypti* (probably carrying a sufficient number of seed viruses) on board sailing ships. Thus, outbreak and extensive vector breeding which ensued in the port communities were inseparable. During a long voyage, those ships functioned as a “miniature urban environment” in the scheme of YFV transmission from one urban environment to another, as shown in Figure 2A. One unique factor of the “miniature urban environment” was the reinforced increase in YFV and vectors on board ships, which likely occurred because of multiple stopovers at YF-infested ports in the tropics during the long journey.

In most historic accounts, the arrival of only one vessel implicated in the disastrous outbreak is prominently mentioned. However, that style of recording is somewhat deceptive, because authors of those records were mostly physicians or medical historians but not the business men well knowledgeable about commercial or shipping activities at the ports. Thus, the number of ships involved in an outbreak was not necessarily only one documented in the records. For example, in the 1800 outbreak in Cádiz, Spain, in most accounts only one ship has been incriminated. However, actually 18 vessels arrived at Cádiz from the tropical areas of the Americas [279]. Similarly, in the 1821 outbreak in Barcelona, 20 ships arrived from Havana between 17 and 23 June [280]. Thus, it is possible that the other vessels not implicated or mentioned in the outbreak records nonetheless could have played a role at least by bringing more mosquitoes and/or seed viruses and intensifying breeding in the busy ports.

Quarantine at the time required inspection of ships from tropical areas. However, legislating quarantine requirement was one thing and enforcing it was another matter. The efficacy of quarantine varied greatly, depending on place. As an example, some outbreaks occurred after the arrival of ships that evaded quarantine requirement, as merchants and ship captains wanted to avoid inspection at all costs. This was observed in the Americas, Europe and Africa. In violation of quarantine law, in some places, a clean bill of health was somehow issued by the authority without actual inspection, suggesting the involvement of corrupt officials. If inspected, the captain’s purposeful denial of any YF cases during voyage was not uncommon, so as not to jeopardize their business. In other cases, wealthy passengers were exempted from quarantine and nearly all asymptomatic travelers were allowed to move on without even a cursory medical examination [281]. In Sénégal, ship captains transferred offshore all passengers to small boats before entering ports, to avoid quarantine [196].

In contrast, in the isolated, imported YF cases documented in temperate cities in modern periods, thus far, infected passengers (mostly a few or less) arriving by jets were almost exclusively residents of the countries in Europe or Noth America free of the vector [282,283]. In addition, their return home was not accompanied with the vectors carrying seed viruses as in the old days. As far as hypothetical, imported case in tropical Asia infested by *Ae. aegypti* is concerned, imported case may increase in the future, as shown by the examples in China in 2016. However, it is another matter whether or not an autochthonous transmission derived from the imported cases would ensue there, because imported cases by jets are similarly not accompanied with an arrival of an army of infected vectors carrying seed viruses. In other words, the chain of the urban transmission cycle depicted in Figure 2A is interrupted in travel by jet. Furthermore, at most international airports, thermal monitoring of arriving passengers has become a routine practice. Though it is not perfect, some YF patients may be detected and segregated at the airport, as happened in 2019 in China, where a traveler from Ethiopia arrived with a febrile condition as a consequence of YF vaccination administered shortly before departure [284].

## 9. Selected Issues with Laboratory Research

Laboratory experiments are indispensable to complement field research. Unfortunately, the use of NHPs is expensive. Furthermore, some key species (such as howler monkeys) do not adapt to confined conditions for experimentation or breeding is difficult. Reconfirmation of the YFV susceptibility of Barbary monkeys in northern Africa (the only YF-susceptible monkey in Africa) is important but currently not possible because of the need for conservation due to the dwindling size of the population. Colonizing important vectors (such as *Haemagogus* spp.) is also difficult, although a progress (still imperfect) was reported recently [285].

### 9.1. Virus Isolation

As shown in Section 4.4, the popular application of vRNA detection in the determination of viremia is problematic if a parallel infectious titer assay is not performed. Since vRNA detection has become a highly popular test in diagnostics, a new term (such as “RNAcidemia”) must be coined and accepted internationally. Because of its popularity, the traditional concept of virus isolation as a gold standard in diagnostics has been much less frequently recognized or forgotten. In one report, urine samples of YF patients were found to be more useful because of the following two advantages: longer detection of vRNA and a noninvasive sampling method [115]. While detection of vRNA is clearly more convenient for processing a large number of clinical specimens in a short time and for genotyping and phylogenetic studies, studies on the phenotypes of the virus are impossible in the absence of virus isolation. Admittedly, virus isolation is often unsuccessful and more time-consuming, but an isolated virus is a treasure box of invaluable information.

### 9.2. Differences between In Vitro and In Vivo

The fact that the flaviviruses represent a group of viruses with a diverse range of host specificity has attracted an attention of many who generated a large amount of interesting molecular data concerning genomic traits responsible for host or vector range determination. A survey of 10 randomly selected articles since 2009 reveals that these studies have been conducted only in vitro. One of the concerns is shown in (ii) below. Replication of some arboviruses transmitted by *Ae. aegypti* is enhanced by natural co-inhabiting viruses (such as members of the Families *Phenuiviridae* and *Spinareoviridae*); but establishing a simulated condition in mosquito cell cultures is a difficult proposition. The use of human cells for a medical aspect of YF research is absolutely essential. However, the application of human cell cultures to unravel the mechanism of YFV transmission in nature is questionable, since the most relevant hosts for this objective are NHPs, as humans (incidental host) are not required for the survival of the virus. As for the choice of mosquito cell cultures, it has been reported that cloned cell cultures, such as C6/36 of *Ae. albopictus*, is not optimal because of the deficiency in the expression of siRNA against a few arboviruses [286]. Strictly speaking, the use of C6/36 cells for YFV research may also be questioned, because *Ae. albopictus* is not the main vector of this virus. However, depending on the type of experiment, insufficient levels of replication of YFV in nearly all cell lines (including Aag2) derived from *Ae. aegypti* do not offer researchers much choice. Listed below are two interesting examples that remind us to be cautious.

(i)In a comparative experiment to study pathogenicity and viral replication in hepatoma cell lines between the vaccine (17D) strain and the wild type (Asibi strain), apoptosis was induced more readily, and virus replicated more with the former than with the latter strain [287,288].(ii)A combination of the following two reports provides a good example for exercising caution for anyone interested in determining the genetic determinants of the phenotypic traits of arboviruses. Earlier, researchers interested in the host range determinant of Sindbis virus (an alphavirus) developed a mutant that failed to replicate in mosquito cell cultures. Surprisingly, when adult mosquitoes were intrathoracically infected with this mutant, the mutant replicated normally in vivo [289]. A caveat in this episode is that intrathoracic inoculation does not simulate natural viral infection because it bypasses the midgut barrier. Accordingly, interpretation of similar reports based only on in vitro system is complicated [290]. This example serves as a good source of caution to exercise in deducing the virus–vector or virus–host interactions in vivo based only on in vitro results. Similarly, this caution is also applied when one deduces what happens under natural conditions based only on laboratory results.

## 10. Concluding Remarks

This review clearly illustrates that arbovirology, which originates in early YF research, has been and continues to be an evolving science. Unravelling the natural transmission mechanisms of arboviruses entails understanding of the complex interactions of many parameters in any ecosystem. Thus, finding numerous ambiguities, deficiencies, puzzles, unknowns and even contradictions about YF listed in this review is not surprising. As mentioned in Section 3.4, treating ticks just like another group of mosquitoes in the arbovirus concept has been one of the poorly publicized major problems. As a consequence, working hypotheses are generated constantly as necessary. If they gather support from a large number of researchers but few actually attempt to validate them scientifically, some of these popular hypotheses may become unintentionally quasi-established principles over time. The notion of the vertebrate reservoir is a good example. A small number of scientists looked for such reservoirs and/or examined the available data and reported the lack of examples or of scientific evidence. Others emphasized that vectors are the true reservoirs [291]. But for some very puzzling reason, these reports did not change the prevailing understanding. The negative consequence of this example is evidenced by the identification of humans and monkeys as the main reservoirs of YFV in a very important and respected international source of public health information about YF many people worldwide depend on [292]. Also, for such an unusually strong statement without an explanation that YF transmission in transitional environments occurs only in Africa [293], presentation of supporting data for the claim would have benefited the readers to clarify the confusion in the Americas.

As advised by the pioneers of YF research [22,50,148], one has to guard against excessively extrapolating, based on a collection of highly plausible but still scientifically incomplete data or on working hypothesis. More recently, it has become common to find in the literature a definitive statement that *Ae. albopictus* is a vector of YF in Brazil. The virus was indeed isolated from this mosquito in nature; and the vector competence of this vector for YFV has been amply confirmed. However, unlike the unquestionable confirmation of its role in dengue outbreaks in temperate Asia (where *Ae. aegypti* does not occur) by this mosquito alone, the scientific evidence of its role in YF transmission in the tropics has not been obtained yet. Thus, it is important to secure scientifically valid evidence before this assumption becomes well established.

Though the above examples represent the negative side of excessive extrapolation, any form of hypothesis is an essential and integral part of thinking to advance science; and the examples of the benefits are incalculable. The point of emphasis here is a need to obtain scientifically valid data for any hypothesis that gains popularity. One problem today is extensive diversification and specialization of research subjects. As a result, too often transmission mechanisms are viewed only from one or a few specialized angles, when capturing them more broadly from the perspective of the natural history of YFV is more desirable. Accordingly, it is important to evaluate the possibility of YF dispersal in the future not based solely on a few selected parameters but in the context of a multifactorial ecosystem including vectors, NHP populations, abiotic factors and the type of environment.

Meanwhile, new research questions are constantly generated. A few recent topics are briefly presented. The first topic is herd immunity. Its importance in YF control was recognized early by John P. Fox of the RF when he was involved in the early YF 17D vaccine trials in Brazil in 1943. It has been recognized that herd immunity is one of the key factors contributing to the cyclical pattern of arboviral disease outbreaks [294]. Since 2017, a coalition of approximately 50 organizations including the WHO, UNICEF, and Gavi began the EYE (eliminate yellow fever epidemics) campaign. The campaign is important, laudable and timely. However, it is puzzling to realize that eight decades after Fox’s interest in the effectiveness of vaccination and a minimal level of immunity for YF control, we still do not have any reliable figure of herd immunity for humans. Generally, crude estimates range from 60 to 99%, 80% being favored by many. As for the herd immunity in NHPs, we know far less. Moreover, it was determined that there is nearly a 20-fold difference in the estimate of YF prevalence, depending on the method and assumptions selected [295]. On top of this, the population in Africa is projected to nearly double by 2065. Thus, when the campaign is closed in 2026, if the goal is not reliably measurable, it would not be easy to scientifically assess the impact of the campaign.

The second topic concerns vaccine development. In the wake of devastating loss of many NHPs in Brazil in the recent YF outbreak, conservation of NHPs became one of the research topics. As described in Section 4.6, some NHPs in Africa are orally infected by YFV. If the NHPs in the Americas are similarly infected, the development of an oral YF vaccine for wildlife may provide a solution not only to this conservation issue but to controlling epizootics and epidemics, by adopting the method developed for rabies control in which vaccine-impregnated bait is used for foxes and other wildlife [296]. The other topic relates to development for humans of a new YF vaccine that does not depend on cold chain. The French vaccine, which satisfied this requirement and which was used to immunize many million of people in Africa, was unfortunately terminated in 1982 due to the occurrence of encephalitis among some recipients. However, before termination, it was also used effectively during a YF outbreak among native American Indians living in a forest in Panamá. Because the location of the YF-stricken tribe was in the roadless forest without electricity or airport, the 17D vaccine could not be used. On the other hand, the French vaccine proved to be highly effective under the difficult circumstances [297]. Because electric services may not be available or unreliable in many locations or environments in the YF zones, the development of an alternative, new vaccine for emergency use is desirable. It should be recognized that many people in the YF zone in Africa are nomadic. Furthermore, many refugees (for economic or political reasons) are constantly on the move. Here is a need to redevelop that type of vaccine which does not depend on cold chains.

The third topic concerns the impacts of drought on YF outbreak. Recently, it was proposed that drought in the sylvan environments in Brazil triggered a chain reaction leading to the emergence of a large outbreak in 2016–2019 [298]. This interesting conclusion deserves further research, because in the past, increased YF activity coincided with the arrival of rainy seasons that rapidly increased the populations of sylvan vectors. A historical and hydrologic study of the standard precipitation index in the states most severely affected (such as Minas Gerais and São Paulo) indeed confirms that the 2013–2014 drought was of historic proportion [299]. However, drought has happened rather frequently in Brazil, though the regions affected varied from time to time. In several major YF outbreaks, including the largest in history in 1934–1940 in Brazil or the 1949–1958 outbreak in Central America, outbreaks were not preceded by a major or prolonged drought. As of 2023, the magnitude of the current drought in the Amazon and Panama is again historic. If the above report of the drought–YF outbreak relation is correct, should we expect another major outbreak again? The drought problem is also common in the YF zone in Africa. As an example, the Democratic Republic of Congo used to be the largest source of fresh water resources in Africa. And yet, the DRC is now suffering from a drinking water crisis for a variety of reasons including drought. When this type of problem occurs, typically residents increase the number of water containers, the breeding sources of *Ae. aegypti*. This practice was precisely the major reason why large YF outbreaks occurred frequently in arid Gibraltar, where residents collected rain water in earthenware called “tinajas” [130], and why the magnitude of dengue outbreaks significantly varied among ethnic groups in southeast Asia because of the difference in tradition regarding how to access to or keep water supplies in daily life [300,301]. The impacts of such factors are yet to be determined for YF in Africa.

With so many unanswered questions, it would not surprise anyone that artificial intelligence (AI) has gained popularity. It is important to stress that however powerful AI may be, one prerequisite for generating correct answers in digital epidemiology is good-quality information in the database, even though some AI programs are said to generate totally new ideas independent of the inventory of information stored. Naturally, if the quality of stored information is questionable, so are the answers generated by AI. This logic is also applicable to currently popular machine reading. That is a serious source of problems, because so many questions about the basics of the sylvan transmission of YFV, the mechanism of viral persistence in nature, the absence of YF in Asia, and many other troubling issues remain unanswered because of the paucity of usable data. In this respect, an aphorism by Walter Lipmann, “mosquitoes do not surrender to elephants”, rings the bell [302]. Peter F. Mattingly used “elephants” for the best arsenal of mosquito control methods humans possessed in the difficult battle waged to suppress vector-borne diseases. Here, it is used as a metaphor for AI. That is why the importance of field research must be emphasized to gather more reliable data. For this to happen, far greater support for field research from research institutions, governmental agencies and funding organizations are necessary to enable international consortia of research institutions to conduct an intensive investigation in the field. Such ideal projects are long term and year round, because short-term projects conducted only in a particular season, even if repeated multiple times, are unlikely to yield answers to the difficult problems of virus maintenance and enzootic activity in the field.

Lastly, this re-examination of many old documents, which did not receive much attention in the past, shows that they are actually a rich source of very useful information even today, providing a number of answers to questions on the one hand and generating research questions and new research proposals on the other. The latest discovery of the evidence that male mosquitoes were once blood feeders, as shown in one of the earliest fossilized mosquitoes in amber [303], reminds us of the need to further examine the past history of the evolution of vector-borne transmission by YFV.

## Figures and Tables

**Figure 1 viruses-16-00084-f001:**
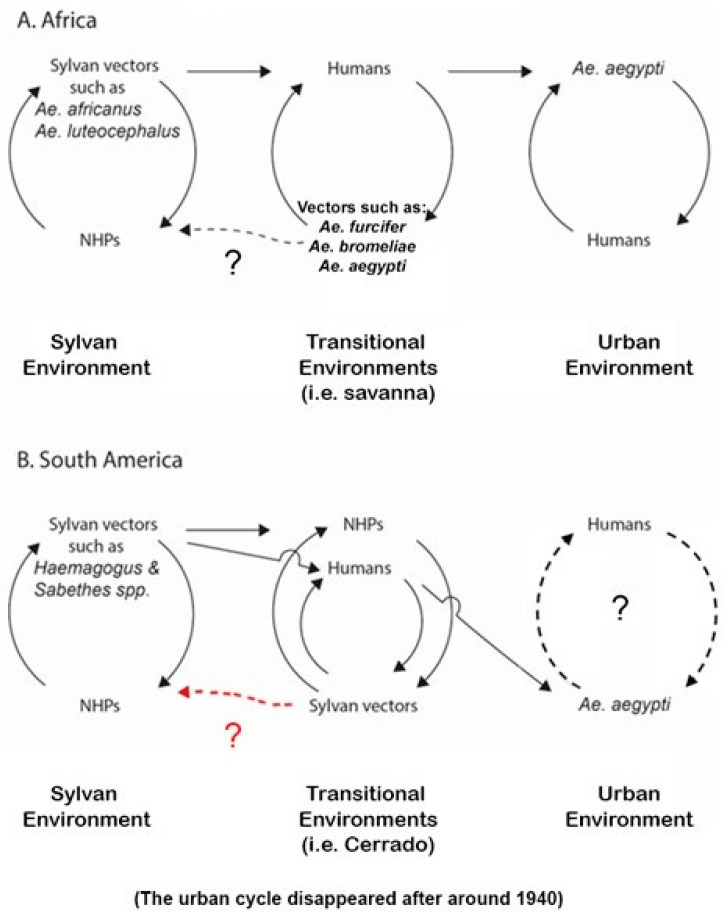
Patterns of YF transmission in Africa and South America based on biological transmission (dotted arrow: possible flow of the virus still unconfirmed).

**Figure 2 viruses-16-00084-f002:**
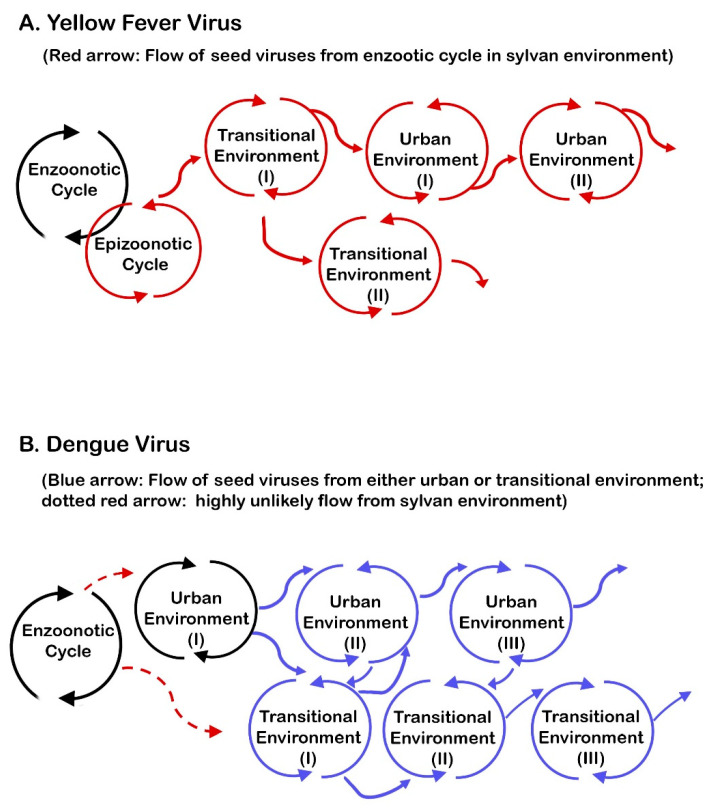
Patterns of the flow of viruses between environments.

**Figure 3 viruses-16-00084-f003:**
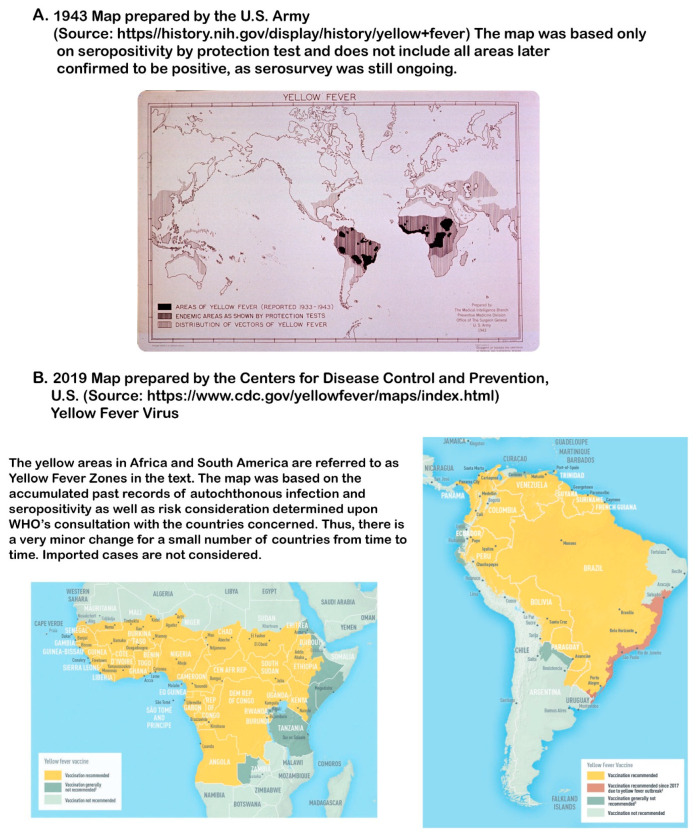
Geographic distribution of YFV.

**Figure 4 viruses-16-00084-f004:**
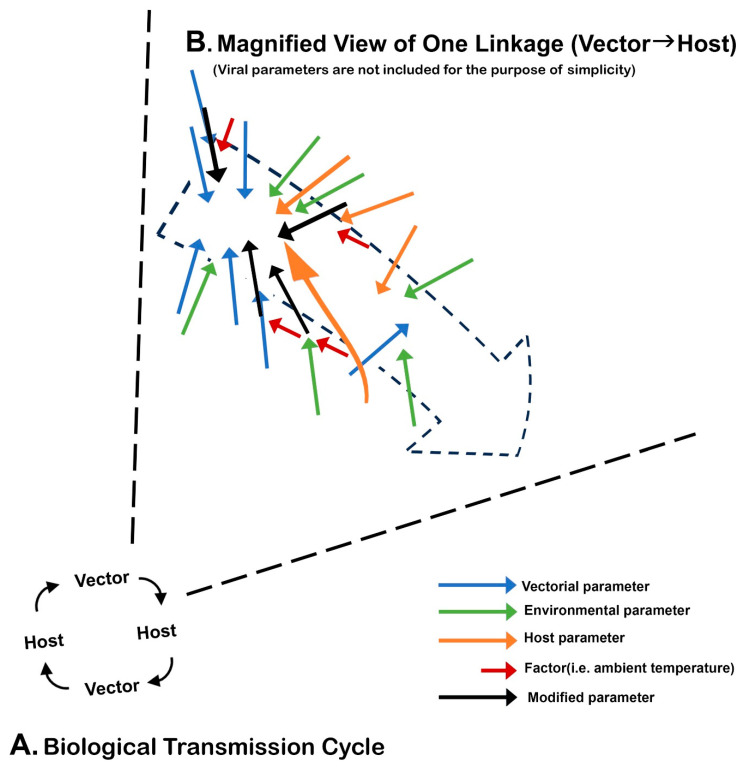
Interactions among parameters and factors in one linkage in a biological transmission cycle.

## Data Availability

Not applicable.

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
