# Peer review of "Mechanisms of Yellow Fever Transmission: Gleaning the Overlooked Records of Importance and Identifying Problems, Puzzles, Serious Issues, Surprises and Research Questions"

_viruses, 2024, doi:10.3390/v16010084_

Round 1

Reviewer 1 Report

Comments and Suggestions for Authors

This review article is just outstanding

I had so much pleasure reading it and will use it extensively

Reviewer 2 Report

Comments and Suggestions for Authors

Yellow fever viruses circulate in complex zoonotic cycles that interface with urban transmission cycles in Africa and historically in tropical America. Important regional differences in zoonotic cycles are known to exist in Africa with regional differences in zoonotic cycles possible in the Americas. The historic failure of yellow fever virus to move out of African zoonotic cycles east to cross the Indian Ocean to infect receptive populations in Asia is a first-class mystery. The need for and design of appropriate research on zoonotic yellow fever in Africa and the Americas should have been the subject of this manuscript.  Instead, the author chose to deliver an encyclopedic history of yellow fever, providing, in addition, anecdotes from other fields. In the place of information, the author has given us words.  Many, many words. The author has shared interesting anecdotes and identified many research questions only partially answered. Mounting a search for a more complete understanding of zoonotic yellow fever is a worthy topic that can be described more parsimoniously.

Comments on the Quality of English Language

There are many awkward and wordy expressions. An experienced editor may be able to help achieve a manuscript telling the same story but in far fewer words. 
